# Photodegradation of Atmospheric Chromophores: Changes in Oxidation State and Photochemical Reactivity

Zhen Mu[a], Qingcai Chen[a*], Lixin Zhang[a], Dongjie Guan[a] and Hao Li[a]

[a] *School of Environmental Science and Engineering, Shaanxi University of Science and Technology, Xi'an 710021, China*

*Corresponding authors:

School of Environmental Science and Engineering, Shaanxi University of Science and

Technology, Weiyang District, Xi'an, Shaanxi, 710021, China.

*(Q. C.) Phone: (+86) 0029-86132765; e-mail: chenqingcai@sust.edu.cn;

**Abstract:** Atmospheric chromophoric organic matter (COM) plays a fundamental role in photochemistry and aerosol aging. However, the effects of photodegradation on chemical components and photochemical reactivity of COM are remain unresolved. Here, we report the potential impacts of photodegradation on carbon content, optical property, fluorophore component, and photochemical reactivity of organic aerosols. After 7 days of photodegradation, fluorescent intensity and absorption coefficient of water-soluble and methanol-soluble COM decrease by 71.4% and 32.0% on average, respectively. Low oxidation humic-like substance (HULIS) is converted into high oxidation HULIS due to photooxidation, the result suggests that the chromophore composition has changed as well as the degree of aerosol aging. COM photodegradation has a significant impact on photochemical reactivity. The generation rate constants of triplet state COM ($^3COM^*$) decrease slightly in ambient particulate matter (ambient PM) but increase in primary organic aerosol (POA) following photodegradation. The results highlight that the opposite effect of photodegradation on photochemical reactivity in POA and ambient PM. The ability of COM generating singlet oxygen ($^1O_2$) decreases obviously, which could be attributed to photodegradation of chromophoric precursors of $^1O_2$. The combination of optical property, chemical component, and reactive oxygen species have an important impact on the atmosphere quality. The new insights on photodegradation of COM in aerosol reinforce the importance of studying DOM related with the photochemistry and aerosol aging.

**Key word:** atmospheric chromophore; photodegradation; EEMs; triplet state; reactive oxygen species.

**TOC:**

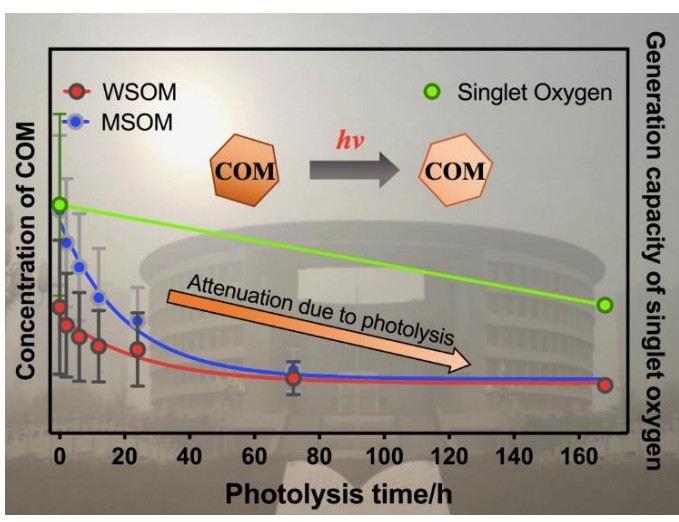

## 1. Introduction

Atmospheric chromophoric organic matter (COM) mainly originates from biomass combustion emission and secondary organic aerosol (SOA) (Andreae & Gelencser, 2006; Budisulistiorini et al., 2017; Graber & Rudich, 2005; Zappoli et al., 1999). Because of the significant absorption for short-wave radiation (Wavelength range from near-ultraviolet light to visible light) (Rosario-Ortiz and Canonica, 2016; Cheng et al., 2016), COM may undergo photochemical processing and have a significant impact on atmospheric components and quality (Zhao et al., 2013; Jo et al., 2016). Therefore, simulation and evaluation of COM photochemistry are necessary for understanding aerosol aging.

As photosensitive substances, the optical properties and components of COM change significantly under solar irradiation (Alkinson et al., 2016; Carlton et al., 2007; Lee et al., 2013; Murphy et al., 2013). On the one hand, optical properties change significantly that due to COM is photo-bleaching in aerosol. Zhong and Jang (2014) reported that mass absorption coefficents (MAC) decreased by 41% on average because wood-burning organic matter (OM) was bleaching, such as conjugated aromatic rings and phenols, and hydroxylated aromatic phenols; Lee et al. (2014) also reported that the MAC of secondary organic aerosol (SOA) continued to decrease in the UV-Vis spectral. On the other hand, photodegradation has a significant effect on the chemical components of COM. COM can be decomposed into small molecules after photodegradation and the photodegraded COM may have lower volatility and higher oxidation degree (Grieshop et al., 2009). COM could also be generated through the photochemical processes, which could be attributed to the formation of SOA. For example, oligomeric COM could be generated by a mixture of anthracene and naphthalene suspensions through self-oxidation under solar irradiation and photo-oxidation of aromatic isoprene oxides were an important source of high molecular weight COM (Altieri et al., 2006; Altieri et al., 2008; Haynes et al., 2019; Holmes and Petrucci, 2006; Perri et al., 2009). SOA may have a more significant light absorption than primary organic aerosol (POA) in the short-wavelength visible and near-UV region (Zhong & Jang, 2014; Saleh et al., 2013; Harrison et al., 2020). As a result, photodegradation plays an important role in the components and properties of COM and thereby change photochemical activity. There are limited studies that comprehensively exploring the characteristics of photodegradation of COM in aerosol.

Photochemical process of COM largely determines the aerosol aging (Mang et al., 2008). On the one hand, COM is often used as reactant in photochemical processes in aerosol. For example, COM could be oxidized by hydroxyl radicals (•OH). The formation of polyols can be attributed to photooxidation of isoprene, which could be initiated by •OH (Claeys et al., 2004; Zhao et al., 2015). Humic-like substance (HULIS) with complex functional groups has significant contribution to photochemistry (George et al., 2015; Nebbioso & Piccolo, 2013; Wenk et al., 2013). On the other hand, COM also participates in atmospheric photochemistry process indirectly through generating reactive intermediates, energy transferring, and involving electron. Upon light exposure, high-energy singlet state COM ($^1$COM*) could be excited. $^1$COM* deactivates by emitting photon

(fluorescence) and intersystem crossing (triplet state ($^3$COM*) generation). $^3$COM* not only can produce photochemical reaction directly, but also can generate reactive oxygen species (ROS), such as singlet oxygen ($^1O_2$), super oxide ($\bullet O_2^-$), and $\bullet$OH, which indicates that $^3$COM* plays a critical role in ROS formation and pollutant attenuation (Paul Hansard et al., 2010; Szymczak & Waite, 1988; Zhang et al., 2014; Rosario-Ortiz and Canonica, 2016; Sharpless, 2012; Haag and Gassman, 1984; Zhou et al., 2019). A lot of COM, such as aromatic ketones (Canonica et al., 2006; Marciniak et al., 1993), benzophenone (Encinas et al., 1985), and phenanthrene (Wawzonek & Laitinen, 1942), have been identified as the precursors of $^3$COM*. Chemical probes, such as 2,4,6-trimethylphenol (TMP) and sorbic acid (SA), are applicable to evaluate the productivity of $^3$COM* (Zhou et al., 2019; Moor et al., 2019; Chen et al., 2021). Compared with $^1$COM*, the characteristics of $^3$COM* are lower formation rate (15~100 times slower than $^1$COM*), lower quenching rate (20000 times lower than $^1$COM*), and higher steady-state concentration (200~1300 times higher than $^1$COM*) (McNeil et al., 2016). Therefore, the reaction rate constant of $^3$COM* is used in evaluating the photochemical reactivity. Considering the potential effect of ROS on aerosol aging and atmospheric quality, it is necessary to clarify the path and mechanism.

COM photochemistry may dominate the chemical composition and the aerosol aging process. In order to illustrate the properties of COM photodegradation and the effect of COM photodegradation on aerosol aging, we simulate the process of COM photodegradation and COM generating ROS in the laboratory. The objectives of the study are (1) to clarify the variation characteristics of carbon content during the COM photodegradation process, (2) to explore the effects of photodegradation on fluorophores and optical properties in water-soluble and methanol-soluble COM, and (3) to investigate the effect of COM photodegradation on photochemical reactivity (photochemical reactivity is characterized by the generation capacity of triplet state and singlet oxygen).

## 2. Experimental Section

### 2.1 Sample Collection

A total of 16 samples were collected (The details of the samples are shown in **Table S1** of SI). The ambient PM samples were collected in Shaanxi University of Science and Technology, Xi'an, Shaanxi Province (N34°22′35.07″, E108°58′34.58″; the altitude of sampling location was about 30 m). The ambient PM samples were collected on quartz fiber filter (Pall life sciences, America) by an intelligent large-flow sampler (Xintuo XT-1025, China) with a sampling time of 23 h 30 min and a sampling flow rate of 1000 L/min. The ambient PM samples were stored in the refrigerator at -20 ℃ prior to use.

The POA samples were collected through a combustion chamber. Straw and coal burning were the main way of heating and cooking in the rural areas in China. Therefore, wheat straw-, corn straw-, rice straw-, and wood-burning samples were collected (Schematic diagram of combustion chamber is shown in **Figure S1**). The POA samples were stored in the refrigerator at -20 ℃ prior to use.

*2.2 Photodegradation experiment*

A quartz reactor was designed for photodegradation experiment (Schematic diagrams of the photochemical devices are shown in **Figure S2**; The detail of the reactor has been described in previous study (Chen et al., 2021)). The photodegradation times were 0 h, 2 h, 6 h, 12 h, 24 h, 3 d and 7 d and a series of photodegraded samples were collected.

*2.3 Carbon content measurement*

The original and photodegraded samples were ultrasonic extracted with ultrapure water (>18.2 MΩ•cm, Hitech, China) and filtered through a 0.45 µm filter (Jinteng, China) to obtain the water-soluble organic matter (WSOM). After water extraction, residual organic matter were further extracted with methanol (HPLC Grade, Fisher Chemical, America) and filtered through a 0.45 µm filter to obtain methanol-soluble organic matter (MSOM). The blank samples were also extracted with the same method.

The measurement method of carbon content has been described previously (Mu et al., 2019). Briefly, 100 µL extract was injected on the baked quartz filter. Then, the wet filter was dried out by a rotary evaporator and the dried filter was analyzed by the OC/EC online analyzer (Model 4, Sunset, America) with the approach of NIOSH 870 protocol (Karanasiou et al., 2015). Organic carbon (OC) was measured in the absence of oxygen. An oven in the instrument was filled with helium and temperature was risen in a gradient style. Different temperatures are needed for particular analysis phases (OC1-310 °C, OC2-472 °C, OC3-615, OC4-850 °C). Element carbon (EC) was measured in the present of oxygen. The oven in the instrument was filled with helium-oxygen gas mixture ($He/O_2$ = 9/1). Different temperatures were also needed for particular analysis phases (EC1-550 °C, EC2-625 °C, EC3-700°C, EC4-775 °C, EC5-850 °C, EC6-870 °C). The products in the heating process were further oxidized to $CO_2$. The carbon content was obtained through the measurement of $CO_2$. Six parallel samples were analyzed and the uncertainty of the method was <3.7% (one standard deviation).

*2.4 Optical analysis*

The light absorption and EEM spectra of WSOM and MSOM were measured by an Aqualog fluorescence spectrophotometer (Horiba Scientific, America). The range of excitation wavelength was 200-600 nm with an interval of 5 nm. The range of emission wavelength was 250-800 nm. The integration time was 0.5 s. The absorption spectra was also recorded in the wavelength range of 200-600 nm. Water and methanol blank samples were measured using the same method and the blank value was subtracted from the sample value. The extracts were diluted to reduce internal filtration effect (The concentrations and dilution factors are shown in **Table S2** of SI).

The EEM data was analyzed by parallel factor analysis model (PARAFAC) to identify fluorophores (The model referred to the previous papers (Murphy et al., 2013; Chen et al., 2016a; Chen et al., 2016b)). WSOM and MSOM (111 samples) were combined in the dataset to create the PARAFC model. According to the EEM characteristics and the residual error variation trend of the

2-7 component PARAFAC models, 4 fluorescent components were identified (Error analysis of the
models is shown in **Figure S4** of SI).

*2.5 Triplet state generation experiment*

As short-lived reactive intermediates, $^3COM^*$ has an important impact on photochemical
process in atmospheric environment (Kaur et al., 2018). Therefore, changes in $^3COM^*$ generation
ability before and after photodegradation were studied. The samples with the photodegradation time
of 0 and 7 d were defined as the original and photolyzed samples, respectively. Only WSOM of
original and photolyzed samples was used in the triplet state generation experiment. A capsule
(**Figure S2(c)**) was designed for this experiment. TMP was used as the capturing agent for the
$^3COM^*$. 60 µL of WSOM extract (Carbon content is shown in **Table S3**) and 60 µL of TMP solution
($c_{TMP} = 20$ µM, Aladdin, China) were mixed in the capsule. The capsule was placed in the reactor
(**Figure S2(a)**). The times of optical excitation were 0, 5, 10, 15, 30, 45, 60 and 90 min, respectively.
Then 90 µL mixed solution was taken out from the capsule at different time points and 30 µL of
phenol solution ($c_{phenol} = 50$ µM, Aladdin, China) was added into the mixed solution (Phenol
solution was used as the internal standard substance for TMP quantification).
TMP was measured by liquid chromatography (LC). The method was as follows: C18 column
(Xuanmei, China); mobile phase: acetonitrile/water = 1/1 (v/v); flow rate: 1 mL/min; UV detector:
detection wavelength 210 nm. The retention time was 14.5 min. Kaur & Anastasio (2018) and
Richards-Henderson et al. (2015) have reported that TMP consumption conformed to first-order
kinetics. The curvy fitting was performed by exponential function among the TMP concentration
($c_{TMP}$/µM), the optical excitation time ($t$/min) and triplet state generation rate constant ($k_{TMP}$/min$^{-1}$):

$$c_{TMP} = a \cdot e^{-t \times k_{TMP}} \tag{1}$$

*2.6 Singlet oxygen generation experiment*

The effect of COM photodegradation on singlet oxygen was studied. A capsule (**Figure S2(b)**)
was designed for $^1O_2$ generation experiment. Only WSOM of original and photolyzed was used in
the singlet oxygen generation experiment. 4-Hydroxy-2, 2, 6, 6-tetramethylpiperidine (TEMP,
$c_{TEMP}$=240 mM, Aladdin, China) was used as the capturing agent for $^1O_2$ and $^1O_2$ was measured by
EPR spectrometer (MS5000, Freiberg, Germany). SA ($c_{SA}$=133.3 µM, Aladdin, China) was used as
quenching agent for $^3COM^*$. The method was as follows: (1) $^1O_2$ was measured before optical
excitation. 40 µL WSOM, 40 µL TEMP, and 40 µL ultra-pure water were mixed in the capsule
(**Figure S2(b)**). Then, 50 µL of the mixed solution was taken out by capillary for EPR analysis; (2)
$^1O_2$ was measured without optical excitation after 60 min. 40 µL WSOM, 40 µL TEMP and 40 µL
ultra-pure water were mixed in the capsule. The capsule was placed in the reactor for 60 min without
illumination. Then 50 µL mixed solution was taken out by capillary for EPR analysis; (3) $^1O_2$ was
measured after 60 min of optical excitation. 40 µL WSOM, 40 µL TEMP and 40 µL ultra-pure
water were mixed in the capsule. The capsule was illuminated in the reactor for 60 min. 50 µL
mixed solution was taken out by capillary for EPR analysis; (4) $^3COM^*$ was quenched and $^1O_2$ was
measured after 60 min of optical excitation. 40 µL WSOM, 40 µL TEMP and 40 µL SA solution
were mixed in capsule. The mixed sample was illuminated in the reactor for 60 min, then 50 µL
mixed solution was taken out by capillary for EPR analysis.

## 3. Results and discussion

*3.1 Effect of COM photodegradation on carbon content*

**Figure 1** describes the changes in carbon content before and after COM photodegradation. In
POA (**Figure 1(A)**), water-soluble and methanol-soluble organic carbon (WSOC and MSOC)
decrease by 22.1% and 3.5%, respectively. The results suggest that WSOC tend to be photodegraded
in POA. In WSOC, the proportion of OC1 (OC1 and OC2-4 are the different stage in the process of
thermal-optical analysis) decreases significantly, which is the main loss of OC. OC1 are
characterized by small molecular weight and highly volatile (Karanasiou et al., 2015) and OC with
these characteristics tend to be photodegraded. In MSOC, there is a process of OC1 translating into
pyrolysis carbon (OPC). The proportion of OPC in MSOC shows an increasing trend (an average
increase of 2.4 times). Pyrolysis carbon is identified as oxygen-containing organic substance. Thus,
the increasing oxygen-containing organic matter may be due to the photo-inducing oxidation
reaction.
POA is fresh and ambient PM has undergone long-term aerosol aging. In ambient PM (**Figure
1(B)**), WSOC is nearly unchanged and MSOC decreases by 18.2%, which is in contrast to POA.
The results reflect that OM has been photodegraded adequately following the photodegradation and
mineralization process in WSOM of ambient PM. However, MSOC with high molecular weight
could not be photodegraded adequately and thereby continue to be photodegraded in laboratory.
The proportions of OC1, OC2-4, and OPC are relatively stable in ambient PM, which indicates that
the decreasing proportions in the different stages are similar and the tendency is also in contrast to
POA. The result reflects that different molecular weight OM may have the similar abilities of
photodegradation in ambient PM. The proportion of different molecular weight OM is nearly
unchanged following the photodegradation in ambient PM.

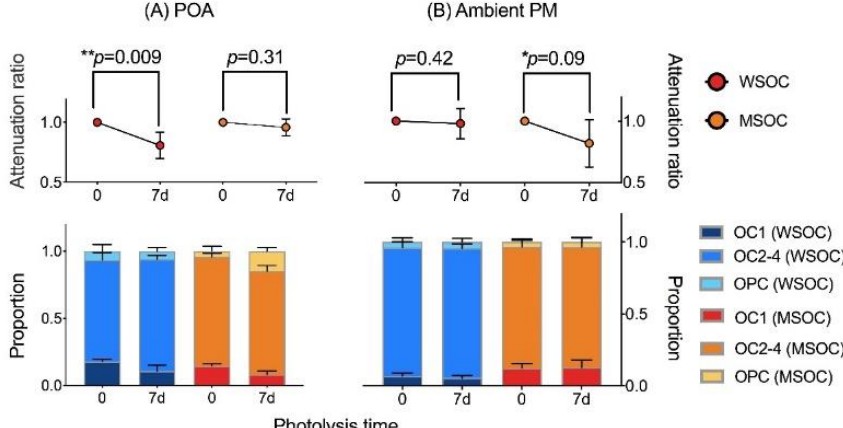

**Figure 1** Changes in carbon content before and after photodegradation. The *p*-value is the probability that two sets
of data have the same level (two-tailed test). * and ** are the significant difference at the 0.1 and 0.01 levels,
respectively.

 *3.2 Effect of COM photodegradation on optical properties*

As shown in **Figure 2**, both absorption coefficient and total fluorescence volume (TFV, RU-
$nm^2/m^3$) significantly decrease following photodegradation, which suggests that COM is photo-
bleaching (Aiona et al., 2018; Duarte et al., 2005; Liu et al., 2016). The attenuations of fluorescence
intensity and absorption coefficient are fitted to first-order decay. The absorption coefficient
decreases by 32.0% and TFV decreases by 71.4% on average. However, as shown in **Figure 3**,
fluorescence intensities increase and decrease in different regions of EEMs (Aiona et al., 2018;
Timko et al., 2015).

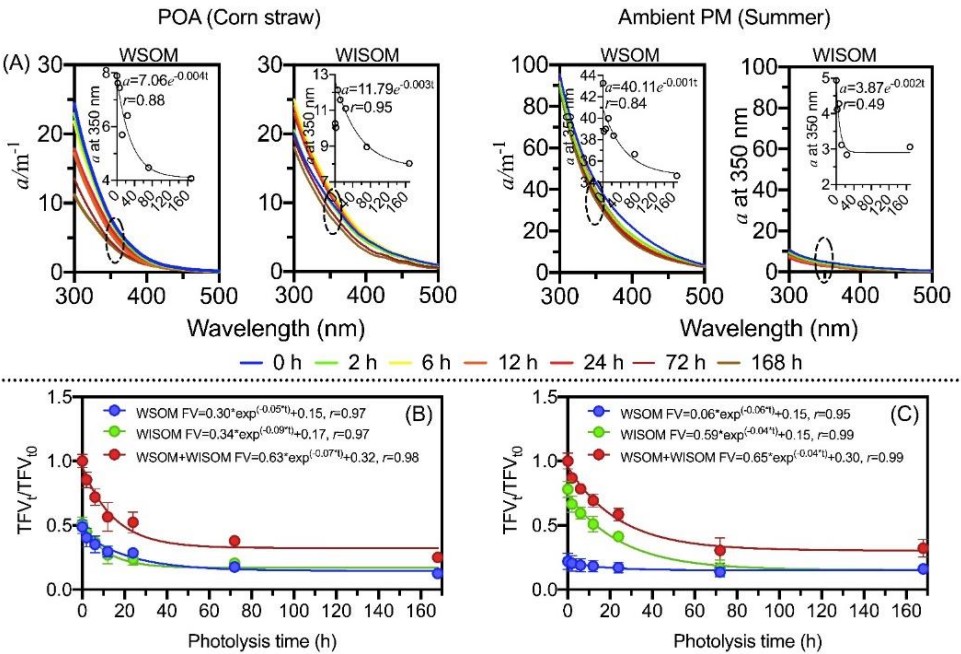

**Figure 2** Changes in absorption coefficient and fluorescence volume during the photodegradation process. (A)
Absorption coefficient. The scatter plot is absorption coefficient at 350 nm. (B) and (C) the attenuation curve of
fluorescence volume in POA (except for the wood sample) and ambient PM, respectively.
In POA (**Figure 2(B)**), TFV decreases by 74.8% on average and the exponential curve method
was used to analysis the attenuation of fluorescence intensity. The attenuation of TFV are significant
similarities between WSOM and MSOM. However, wood-burning samples are distinct from other
POA samples, TFV of wood-burning COM only decreases by 9.0% and fluorescence volume of
MSOM of wood-burning samples remain almost unchanged (**Figure S7**). There are two main
reasons. On the one hand, methanol-soluble secondary OM is generated slightly in wood-burning
samples (Zhong & Jang, 2014). On the other hand, methanol-soluble wood-burning COM is
difficultly photodegraded. In addition, fluorophores photodegradation also depends on the
photochemical environment, such as solution pH (Aiona et al., 2018), salinity (Xu et al., 2020), and
temperature (Yang et al., 2021). Therefore, we suppose that photodegradation of wood-burning
COM may largely depend on photodegradation environment.

The attenuation rate constant of TFV in ambient PM ($k = 0.04$ h$^{-1}$) is lower than that in POA

($k = 0.07$ h$^{-1}$). In ambient PM, TFV of MSOM decreases by 79.4% but WSOM decreases by 26.7%
(**Figure 2(C)**). The attenuation of TFV and carbon content is identical with each other. The results
suggest that MSOM has greater ability to be photodegraded than WSOM in ambient PM. It is worth
noting that 72 h could be considered as the end point of aerosol photo-aging because TFV maintains
a constant value after 72 h both in POA and ambient PM.

COM can be decomposed and transformed due to photodegradation in aerosol (Wong et al.,

2015). Fluorophores are studied by the approach of EEMs-PARAFAC. Although previous study
has analyzed the water-soluble and methanol-soluble fluorophores separately (Tang et al., 2020a),
based on the Chen's studies (2020; 2016b), water-soluble and methanol-soluble samples were
combined to create the PARAFAC model to illustrate the distribution of fluorophores in WSOM
and MSOM and solvent had no significant effect on the EEMs of complex mixtures in aerosol. As
shown in **Figure 3(A)**, four fluorophores are identified. The fluorescence peaks of C1 and C2 appear
at (Ex./Em. = 224/434 nm) and (Ex./Em. = 245/402 nm). The peaks are similar to high and low
oxidation HULIS, respectively (Chen et al., 2016b; Birdwell and Engel, 2010). The peaks of C3 and
C4 appear at (Ex./Em. = 220/354 nm) and (Ex./Em. = 277/329 nm) and these two fluorophores are
associated with protein-like organic matter (PLOM-1 and PLOM-2) (Sierra et al., 2005; Huguet et
al., 2009; Chen et al., 2016a and 2016b; Coble, 2007; Fellman et al., 2009).

The content of fluorophores changes significantly during the photodegradation process. In

POA (**Figure 3(B)**), the relative content of high-oxidation HULIS (C1) increases by 63.0% and the
relative content of low oxidation HULIS (C2) decreases by 88.0% in WSOM. Changes in proportion
indicate that high-oxidation HULIS fluorophore (C1) could be generated and low oxidation HULIS
(C2) may be converted into high oxidation HULIS (C1) due to photooxidation (Tang et al., 2020b;
Chen et al., 2020). PLOM (C3&C4) decreases 19.7%, which indicates PLOM (C3&C4) can be
photodegraded. In MSOM of POA, no regularity of variation is found in low oxidation HULIS (C2)
and PLOM (C3&C4) and the content of high-oxidation HULIS (C1) increases by 17.5%, which can
be attributed to photo-mediated secondary reaction.

The contents of PLOM (C3&C4) in ambient PM (19.4%) (**Figure 3(C)**) are significantly lower

than that in POA (43.3%). The content of high-oxidation HULIS (C1) multiplied 6.9 times and the
low-oxidation HULIS (C2) decreases by 40.2% in WSOM, the variation is similar to POA. Thus,
high-oxidation HULIS could be used to trace the degree of aerosols photo-aging. In MSOM of
ambient PM, the content of high-oxidation HULIS (C1) increases by 43.5% and no regularity of
variation is found in low oxidation HULIS (C2) and PLOM (C3&C4).

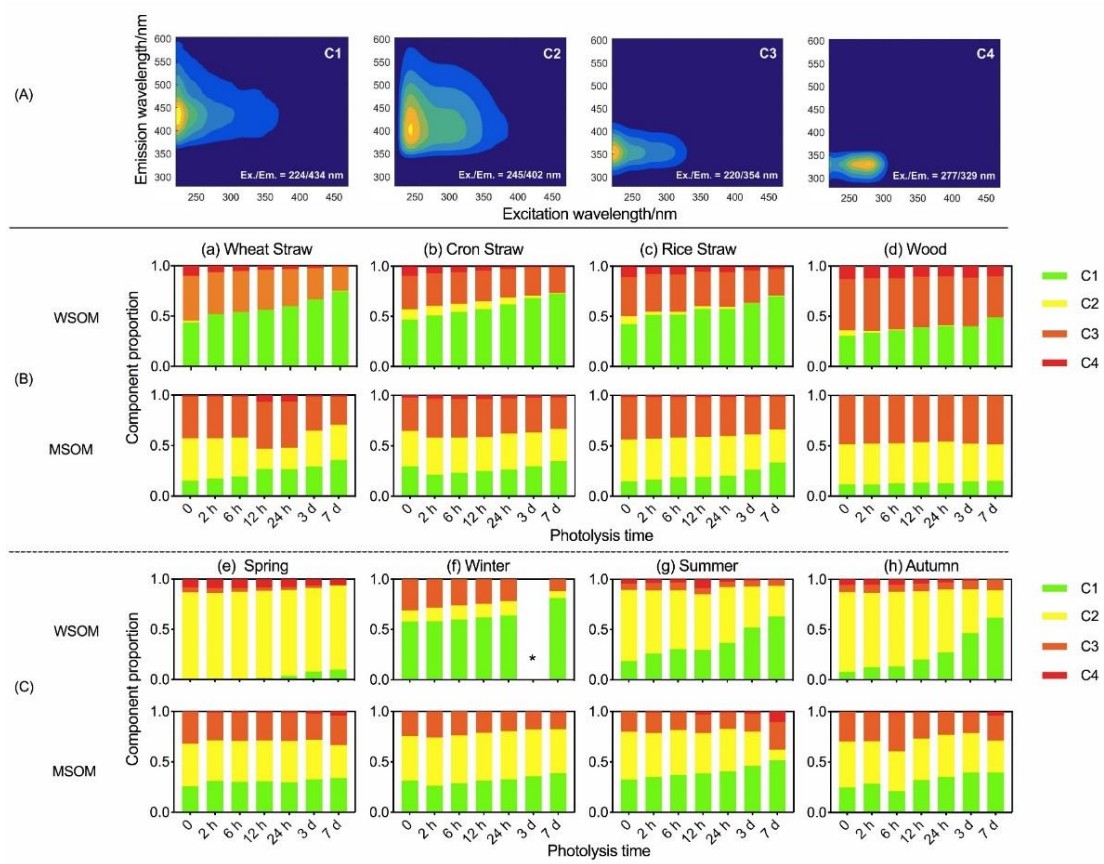


**Figure 3** (A) EEM spectra of fluorophores; (B) Changes in proportion of fluorophores in POA; (C) Changes in
proportion of fluorophores in ambient PM. *: The data of 3-day photolysis of water-soluble fluorophores in winter
is unavailable.
*3.3 Effect of COM photodegradation on aerosol photochemical reactivity*

COM photodegradation has a significant effect on photochemical reactivity of COM in aerosol.

The photochemical activity is quantitative analyzed by the yield of $^3COM^*$ and $^1O_2$. Only WSOM
of original and photolyzed samples was used to measure the yield of $^3COM^*$ (Original samples with
photodegradation time of 0, photolyzed samples with photodegradation time of 7d; details of
samples are described in section 2.2). **Figure 4** shows the variation of triplet state generation before
and after the photodegradation (Consumption curves of TMP are shown in **Figure S8**). In ambient
PM, compared with original samples, the generation rate of triplet state decreases by 11% on
average in photolyzed samples, while statistical analysis shows that the changes are not obvious ($p$
$= 0.38$, two-tailed test). On the contrary, in POA, photodegradation promotes triplet state generation
significantly, the triplet state generation rate increases by 75% on average in photolyzed samples.
($p = 0.07$, two-tailed test). The results that triplet state generation remains unchanged or increases
in different aerosols following photodegradation are unexpected and can be explained by recent
study (Chen et al. 2021): on the one hand, only a small proportion of water-soluble OM could

generate triplet state in aerosol and fluorophores do not represent the OM with the ability to generate triplet state. Therefore, triplet state generation could not be evaluated only by fluorescence intensity. On the other hand, we use a high concentration of TMP, in this case, TMP mainly captures high-energy triplet state (Rosado-Lausell et al., 2013; Chen et al., 2021). Thus, COM, that could generate a high-energy triplet state, may not be photodegraded in ambient PM.

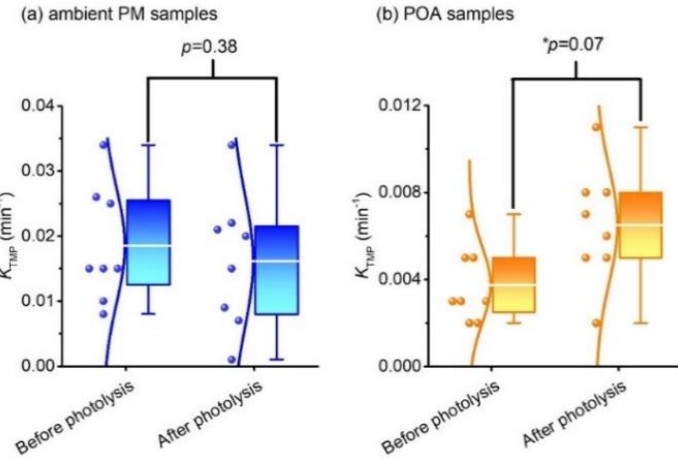

**Figure 4** Changes in the triplet state generation. (a) Ambient PM; (b) POA. The line from bottom to top in the box plots are minimum, first quartile, the average value (white lines), third quartile, and maximum, respectively. The $p$-value is the probability that two sets of data have the same level (two-tailed test). * represents a significant difference at the 0.1 level.

COM can generate triplet state and further generate singlet oxygen (McNeill and Canonica, 2016). The effect of COM photodegradation on singlet oxygen is studied. WSOM of original and photolyzed samples was used to measure the yield of $^1O_2$ (EPR spectra of all samples is shown in **Figure S9** and **Figure S10**). As shown in **Figure 5**, a decrease in the yield of $^1O_2$ reveals the inhibiting effect of COM photodegradation on photochemical activity both in ambient PM and POA. In POA, as shown in **Figure 5(A)**, (I) Before optical excitation, there is little $^1O_2$ both in original and photolyzed samples; (II) After 60 min in dark, $^1O_2$ are generated both in original and photolyzed samples, which suggests POA could generate $^1O_2$ without optical excitation. The content of $^1O_2$ in original samples is higher than that in photolyzed samples; (III) After 60 minutes of optical excitation, as expected, compared with the samples without optical excitation, the content of $^1O_2$ increases by 3 times both in original and photolyzed samples. The content of $^1O_2$ in original samples is also higher than that in photolyzed samples (42.1%), which prove the inhibiting effect of COM photodegradation on $^1O_2$ generation; (IV) However, the content of $^1O_2$ is nearly unchanged when the triplet state is quenched by sorbic acid. Therefore, the results indicate that the low-energy $^3COM^*$ ($E_T <239$ kJ/mol) may be the main precursor for $^1O_2$ ($E_T = 94$ kJ/mol) in POA, because sorbic acid is a capturing agent for high-energy triplet state (triplet energies $E_T = 239\text{-}247$ kJ/mol) (Zhou et al., 2019; Moor et al., 2019). In addition, COM photodegradation does not change the mechanism of low-energy $^3COM^*$ generating $^1O_2$ in POA.

In ambient PM, as shown in **Figure 5(B),** (V) Before optical excitation, the content of $^1O_2$ is very low in original and photolyzed samples, which is similar to POA; (VI) Compared with (V), the content of $^1O_2$ is almost unchanged after 60 min in dark, which is different from POA. The result suggests ambient PM could not generate $^1O_2$ without optical excitation. (VII) After 60 minutes of optical excitation, the content of $^1O_2$ increases significantly and the content of $^1O_2$ in original samples is higher than that in photolyzed samples (41.0% higher). (VIII) When the triplet state is quenched by sorbic acid, $^1O_2$ does not be generated. The result suggests that the precursor of $^1O_2$ is quenched and $^1O_2$ is mainly generated by high-energy $^3COM^*$ in ambient PM. COM with the ability of generating high-energy triplet state could be photodegraded, which directly leads to the decrease in $^1O_2$ in ambient PM. The quenching effects of sorbic acid on triplet state in POA and ambient PM are different because of the different energy of triplet state.

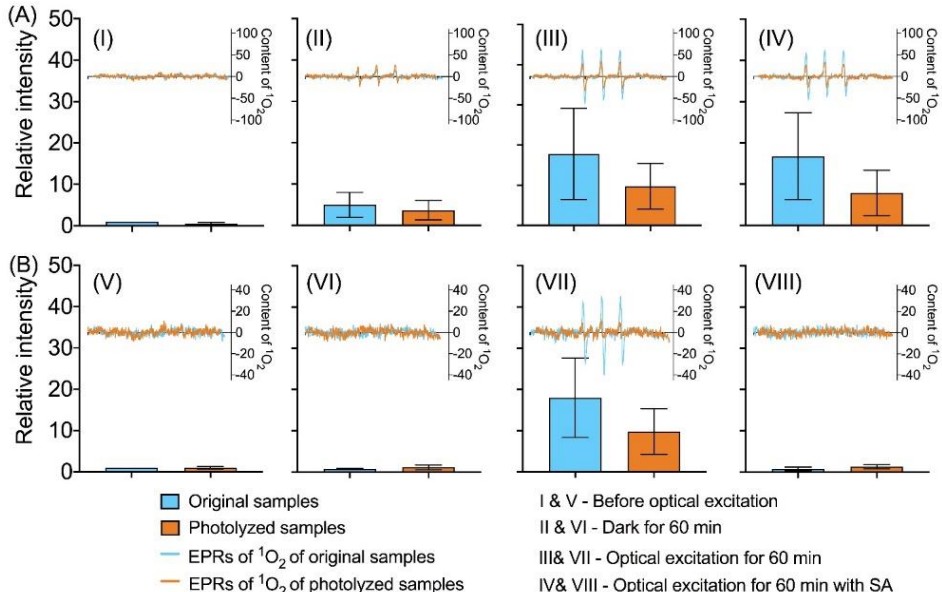

**Figure 5** Changes in COM generating $^1O_2$ before and after photodegradation. (A) POA; (B) Ambient POA.

## 4. Implication

We made a comprehensive study in COM photodegradation and the effect of COM photodegradation on optical property, chemical component, and photochemical reactivity to reveal the characteristics of COM photodegradation. COM photodegradation could result in reduction of carbon content, attenuation of optical property, and change in fluorescent component. We also proposed that COM photodegradation should be evaluated from three aspects for further study. (1) The impact of COM photodegradation on carbon content was unclear. Previous studies have revealed that WSOC did not significantly change in the river DOM (Gonsior et al., 2009) and 0.2% of DOC was mineralized (Tranvik et al., 1998). However, the observation in the study suggested that changes in carbon content were different in aerosols, which could be attributed to the differences in original components. (2) Attenuation in optical properties was significant. Absorption coefficient and fluorescence intensity could be thought of as a tracer for molecular weight (Stewart

& Wetzel, 1980). Therefore, optical properties indicated the changes in molecular weight of COM during the photodegradation process. The characteristic could be suitable for exploring the impact of photodegradation on COM components. (3) COM Photodegradation may dominate the fluorophores components (Aiona et al., 2018; Timko et al., 2015). High molecular weight COM could be decomposed into low molecular weight COM during the photodegradation process. The conversion of low-oxidation HULIS to high-oxidation HULIS was observed. Changes in COM may represent the oxidation degree of organic substances. Therefore, we suggested that optical parameter and oxidation degree of organic molecules should be use for characterizing the aerosol photo-aging process (Maizel et al., 2017).

Photodegradation could not only change the properties and components of COM, but also change their photochemical reactivity, which further had a potential impact on the aerosol fate. Photodegradation and/or conversion of COM could be considered to be the main influence factor for photochemical reaction capacity (McNight et al., 2001; Zepp et al., 1985). Photochemical reactivity was quantified by the yield of triplet state and $^1O_2$ in our study. However, two different methods, two different results. COM photodegradation could restrain $^1O_2$ generation but the effect of photodegradation on $^3COM^*$ was unclear. Photodegradation had a significant inhibiting effect on the $^1O_2$ yield in aerosols (Latch et al. 2006; Chen et al., 2018). We insisted that aerosol aging would be changed by photodegradation due to the yield of $^1O_2$ was changed. Changes in triplet state generation were uncertain in ambient PM and POA. There were two reasons for this. On the one hand, only a small amount of COM was the precursor of $^3COM^*$ in aerosols. On the other hand, the energy of capturing agents was closely related to $^3COM^*$ quantification and thereby $^3COM^*$ could not be captured completely. Other capturing agents may lead to different results. Thus, $^3COM^*$ could not properly illustrate photodegradation. COM photodegradation could play an important role in the content of ROS and ROS could calibrate the COM photooxidation (Claeys et al., 2004). Given the results, the interaction effect was significant in aerosol.

In summary, atmospheric photochemistry process had a remarkable impact on aerosol aging. Prediction of atmospheric lifetime and improvement of quality were strongly associated with photochemistry. We proved that carbon content, absorption coefficient, fluorescence intensity, and photochemical reactivity were useful to reflect COM photodegradation process and aerosol fate. In addition, COM photodegradation had a different impact on chemical reactivity in different aerosols, which may have different mechanisms. Therefore, the mechanisms of COM photodegradation affecting aerosol photo-aging deserved further investigation.

**Data availability.** All data that support the findings of this study are available in this article and its Supplement or from the corresponding author on request.

**Supporting information.** Additional details, including Tables S1−S5, Figures S1−S10, calculation of optical characteristics of WSOM/WISOM, are contained in the SI.

**Author contributions.** QC and ZM designed the experiments and data analysis. ZM and LZ performed sample collection. ZM performed the photochemical experiment. ZM and DG performed the OC/EC analysis and optical analysis. HL performed the EPR analysis. QC prepared the paper with the contributions from all co-authors.

**Competing interests.** The authors declare that they have no conflict of interest.

**Acknowledgments.** We thank the National Natural Resources Foundation for its financial support.

**Financial support.** This work was supported by the National Natural Science Foundation of China (grant numbers 41877354 and 41703102), and the Youth Science and Technology Nova Program of Shaanxi Province (2021KJXX-36).

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
