# Peer review of "Photodegradation of Atmospheric Chromophores: Changes in Oxidation State and Photochemical Reactivity"

_Atmospheric Chemistry and Physics, 2020_

## Referee Comment (RC1) · Anonymous Referee #1 · 22 Dec 2020

The manuscript "Photo-degradation of atmospheric chromophores: type conversion and changes in photochemical reactivity" addressed the photochemical degradation of atmospheric COM and a loss of this material of 70% within 7 days of light exposure. The involvement of reactive oxygen species also was addressed. Grammar should be checked and grammatical errors are distracting in reading the manuscript. The presented data does not add a lot of additional information on the photochemical behavior of aerosols. Previous studies described in detail the photochemical kinetics on more relevant SOA. It is not clear to me what the novelty of this paper is and perhaps the authors need to strengthen and highlight better the novel contribution. It also was not helpful to see limited details in the method sections. More specifically, it is not clear to

me how the aerosols generated here in the lab resembles similarity to once in the atmosphere. An explanation is needed why the combustion of wheat straw, corn straw, rice straw and wood was chosen to generate aerosols. To elaborate on my assessment, please see more specific comments below:

Actinometry needs to be undertaken to confirm the dose of light in the reaction chamber, otherwise it is not clear what the exposure was according to the geometry and pathlength.

Line 128: what was the dilution factor? Line 134: How many samples were used to create the model? Were water soluble and methanolic samples combined in the dataset to create the PARAFC model? How were the EEMs of the methanolic extract measured?

Line 150: Was is isocratic and at what retention time did TMP elute?

Figure 3: The raw absorbance should be converted to apparent absorption coefficient a, so that it is normalized to pathlength.

Line 220: It is very easy to contaminate fluorescence samples that than show protein-like fluorophores. How did the blank samples compare? Was this protein-like fluorescence apparent in controls? This needs to be carefully addressed so that QA/QC can be assessed.

Line 240: Triplet state of what?

Line 270-280: It is well known that singlet oxygen is photochemically generated and very well correlated to UV-vis absorption, so this all seems to me a generic trend that has been described previously in detail, even in SOA. I am also not convinced that an ill-defined and not quantifiable low-energy 3COM* even exists and less so being the main precursor of singlet oxygen, but of course this is debatable.

---

## Referee Comment (RC2) · Anonymous Referee #2 · 24 Dec 2020

The manuscript 'Photodegradation of atmospheric chromophores: type conversion and changes in photochemical reactivity' provides results on the photochemical aging of atmospheric aerosols. The results include OC/EC analysis, parallel factor (PARAFAC) analysis of excitation-emission matrices, and photosensitization of singlet oxygen with each measured as a function of solar irradiation. The results are of interest and merit eventual publication in the atmospheric chemistry literature, however, the novelty, writing, and presentation require significant improvement and does not meet the standard for publication in Atmospheric Chemistry and Physics in its current state. My comments are outlined below.

[Figure]

General comment: The manuscript would benefit greatly from a substantial revision to improve the writing quality (see minor comments for extensive recommended changes), conceptual framework, and referenced literature, to increase the readability of this work.

Major comments: 1) The title of this manuscript could be revised to remove the 'type conversion' terminology which is ambiguous and not commonly used. I would recommend making it more explicit that you are talking about oxidation by revising to: Photodegradation of Atmospheric Chromophores: Changes in Oxidation State and Photochemical Reactivity.

2) In the first paragraph of section 3.2, the results on the decrease in absorbance vs. fluorescence (or TFV) are not well separated and it is extremely difficult to separate when the authors are referring to absorbance or the fluorescence of these. It appears decay constants are provided for TFV but nor for absorbance.... Do the photolysis decay kinetics differ significantly for the chromophores vs. fluorophores? Or does the absorbance decay at the same rate as TFV?

3) What are the estimated atmospheric photolysis lifetimes including the factor of 1.2-1.3 mentioned in Sec. 2.2? Also according to Figure S1 in the supplement, your light source has almost no flux from 300-350 nm where the aerosols sample absorb most strongly and differs substantially from the solar spectrum. This should be addressed at some point in the manuscript.

4) On pages 8-9, the impacts of photolysis on the EEMs and individual components could be discussed in much more detail with comparison to more literature. Here are some additional references on this topic:

Murphy, K. R.; Timko, S. A.; Gonsior, M.; Powers, L. C.; Wünsch, U. J.; Stedmon, C. A. Photochemistry Illuminates Ubiquitous Organic Matter Fluorescence Spectra. Environ. Sci. Technol. 2018, 52 (19), 11243–11250

[Figure]

Harrison, A. W.; Waterson, A. M.; De Bruyn, W. J. Spectroscopic and Photochemical Properties of Secondary Brown Carbon from Aqueous Reactions of Methylglyoxal. ACS Earth Space Chem. 2020, 4 (5), 762–773.

5) In Figure 3A, the absorbances of WSOM and WISOM are nearly identical in the POA samples but differ by a factor of 10 in ambient PM. Is this a genuine difference in optical properties of these fractions i.e. does the water-soluble fraction actually absorb much more strongly than the water-insoluble fraction? This would be in contrast to previous brown carbon literature worth discussing in the manuscript. The use of mass absorption efficiency (MAE) is mentioned in the Supplement but not in the manuscript.

6) Are the WSOM and WISOM combined for the PARAFAC analysis? I would expect that there would be some differences in the fluorophores present in each fraction. See this recent ACP paper where there are clear differences in the water-soluble and methanol-soluble fractions:

J. Tang, et al. "Molecular compositions and optical properties of dissolved brown carbon in biomass burning, coal combustion, and vehicle emission aerosols illuminated by excitation–emission matrix spectroscopy and Fourier transform ion cyclotron resonance mass spectrometry analysis" Atmos. Chem. Phys. 20, 2513-2532, 2020.

7) Tabulation of the relative changes in each of the components in Figure 4B-C would improve the presentation of the results. This is information is all packed into the paragraph starting at Line 222 and is difficult to parse.

8) The paper demonstrates that low energy triplets are the main precursor for singlet oxygen , but do very high energy triplets form singlet oxygen with less efficiency in general? It would seem intuitive to me that when the energy gap between singlet and triplet COM is large, the photosensitization reaction will be less efficient.

9) Implication section is short and lacks significance or contextualization in the large amount of literature on the photochemical oxidation of chromophoric organic matter in

atmospheric aerosols.

Minor comments:

Chromophoric Organic Matters should be 'matter' instead of 'matters'. Consider how making this change will impact verb conjugation

Line 9: Change 'Furtherly' to 'Furthermore'

Line 12: Change 'particle' to 'particulate'

Line 15: Change 'result also enunciate' to 'results also highlight'

Line 30: Change 'improve' to 'is necessary for'

Line 38: Change 'Photochemistry have' to 'Photochemistry has'

Line 101: Change 'time' to 'times'

Line 127: Change 'analyzed' to 'measured'

Line 131: Change 'excitation' to 'emission'

Line 134: Change 'chromophores' to 'fluorophores'

Line 176: Change 'photolysis' to 'photolyze' or rephrase sentence to 'undergo partial photolysis'

Line 179: Change 'result' to 'results'

Line 186: Change 'Contrast with' to 'In contrast to'

Fig. 2b: Heading says 'Ambint PM' rather than 'Ambient PM'

Line194-195: Change 'represent an obvious decreasing trend due to aerosol photolysis' to 'significantly decrease during aerosol photolysis'

Line 197: Change 'subduction' to 'decay'

Line 210: Change 'photolysis' to 'photolyze'

Line 208-210: Sentence that begins with 'The low attenuation result from COM' is unclear and needs revision

Line 219: Change 'chromophores' to 'fluorophores'

Line 220: Change 'identified as' to 'associated with'

Line 243: Change 'show' to 'shows'

Line 243: Change 'do not significant affect' to 'does not significantly affect'

Line 260: Change 'through the approach of' to 'using'

Line 261: Change 'EPRs' to 'EPR'

Line 267: Change 'increase by 3 times' to 'increases by a factor of 3'

Line 277: Line 176: Change 'do' to 'does'

Line 278: Change 'The mechanism is same as' to 'The mechanism is the same as the'

Line 285: Change 'disappear' to 'disappears'

Line 288-289: Consider changing 'restraining' to 'attenuating' or 'inhibiting'. 'Restraining' is an odd word to use for this.

Line 291: Change 'prove' to 'show'

Line 293: Change 'prove' to 'study'

Line 305: Change 'reflect' to 'reflects'

---

## Author Response (AR1)

**Response to reviewers for the manuscript "Photo-degradation of atmospheric chromophores: type conversion and changes in photochemical reactivity" (acp-2020-1223)**

We appreciate the comments from the editor and reviewer. According to the reviewer's comments, we have revised this paper. The details are as follows. *The blue italics are comments of reviews. The red italics are improvements and original text of reviews.* The black font are responses.

**Response to Anonymous Referee #1**

*The manuscript "Photo-degradation of atmospheric chromophores: type conversion and changes in photochemical reactivity" addressed the photochemical degradation of atmospheric COM and a loss of this material of 70% within 7 days of light exposure. The involvement of reactive oxygen species also was addressed. Grammar should be checked and grammatical errors are distracting in reading the manuscript. The presented data does not add a lot of additional information on the photochemical behavior of aerosols. Previous studies described in detail the photochemical kinetics on more relevant SOA. It is not clear to me what the novelty of this paper is and perhaps the authors need to strengthen and highlight better the novel contribution. It also was not helpful to see limited details in the method sections. More specifically, it is not clear to me how the aerosols generated here in the lab resembles similarity to once in the atmosphere. An explanation is needed why the combustion of wheat straw, corn straw, rice straw and wood was chosen to generate aerosols. To elaborate on my assessment, please see more specific comments below:*

We appreciate the comments from reviewer. We appreciate the positive evaluation of this work. According to the reviewer's comments, we have revised this paper. The details are as follows. *The blue italics are comments of reviews. The red italics are improvements and original text of reviews.* The black font are responses.

(1) We studied the characteristics of COM photo-degradation, the potential effects of COM photolysis on the photochemical reactivity, and the contribution of COM to reactive oxygen species (ROS) in aerosol in this paper. The novelty of this paper is the characteristics and mechanisms of COM photolysis and the effect of COM photolysis on aerosol aging. We have corrected the paper to strengthen and highlight better the novel contribution. For example,

We have corrected "*the characteristics of COM photo-degradation and the potential effects of COM photolysis on the photochemical reactivity are illustrated*" to "*Here, we report the characteristics of COM photo-degradation, the potential effects of COM photolysis on the photochemical reactivity, and the contribution of COM to reactive oxygen species (ROS)*" in abstract in the improved paper.

We have corrected "*In order to illustrate the effect of COM photo-degradation on the optical properties and photochemical reactivity in aerosols, we simulate the photolysis process of primary organic aerosol (POA) and ambient particle matter (ambient PM) in laboratory*" to "*The chemical composition and atmospheric quality are significantly affected by aerosol aging. In order to illustrate the properties of COM and the effect of COM photolysis on aerosol aging, we simulate the*

40 *process of COM photolysis and COM inducing ROS in primary organic aerosol (POA) and ambient particulate matter (ambient PM) in laboratory*" in section 1 in the improved paper.

We have added "*We made a comprehensive study of COM photo-degradation, changes in optical properties and chemical compositions, the effect of COM photo-degradation on photochemical activity and aerosol aging. The properties of COM photo-degradation were revealed. COM photo-*
45 *degradation could be explained by reduction of carbonaceous components, decrease of light absorption capacity, and attenuation of fluorescence intensity. There are great differences in various COM in aerosols. Therefore, we suggested that the properties COM photo-degradation could be comprehensively characterized by carbonaceous components and optical characteristics*" in section 4 in the improved paper.

50 We have added "*The effects of COM photo-degradation on the photochemical activity in aerosols are studied. We evaluated the effect of COM photo-degradation on the photochemical activity. The ability of triplet state generation and $^1O_2$ yield was chosen to quantify the photochemical activity*" in section 4 in the improved paper.

(2) We described the generated method of POA in detail. The supplementary and revision of method
55 have been added in the improved paper. For example,
We added the schematic of combustion equipment in improved paper.

[Figure]

*Fig.1 Schematic diagrams of combustion equipment for POA.*

We have corrected "*Wheat straw, corn straw, rice straw and wood were burned at about 500 ℃ in*
60 *the tube stove*" to "*As shown in Fig.1, Wheat straw, corn straw, rice straw and wood were burned in the annular combustion chamber when temperatures rose to 500 ℃*" in section 2.1 in the improved paper.

We have corrected "*A high-purity quartz reactor was designed for the photolysis experiment (Fig.1a). A rubber gasket was embedded on the upper edge of the reactor. The reactor was clamped*
65 *with a high-purity quartz cover to form a sealed environment*" to "*The material of reactor is quartz (Fig.2a) and the reactor was designed for photolysis experiment. The reactor was sealed through clamping a quartz cover to reactor*" in section 2.2 in the improved paper.

We have corrected "*Two vents were designed in the low position of the reactor. The vents were connected to water circulator to ensure that the temperature was about 25°C in the reactor*" to "*Two*

*air vents were used to air exchange and Two water cycle vents were connected to water circulator to ensure that the temperature was about 25°C in the reactor*" in section 2.2 in the improved paper.

(3) Straw and coal burning are the main way of heating in China, especially in the rural areas. Coal burning is also the main source of energy in China. Therefore, the combustion of wheat straw, corn straw, rice straw and wood were chosen to generate aerosols.

We have added "*Straw and coal burning are the main way of heating and cooking in the rural areas in China. Therefore, the combustion of wheat straw, corn straw, rice straw and wood were chosen to generate aerosols*" in section 2.1 in the improved paper.

*1. Actinometry needs to be undertaken to confirm the dose of light in the reaction chamber, otherwise it is not clear what the exposure was according to the geometry and pathlength.*

(1) The light intensity was measured by an optical radiometer (Perfectlight Inc.) and the absorbance can be estimated in the reactor (Fig.2). Each sample was placed in the same position under the same light intensity to maintain the same dose of light.

(2) According to the previous study (Laszakovits et al., 2017), we calculated the dose of light by chemical method.

We have added "*4-nitroanisole (PNA, 10 μM) and pyridine (pyr, 10 mM) were used in the method. Firstly, the absorbance of the mixture of 4-nitroanisole and pyridine was measured. Then, the mixture of 4-nitroanisole and pyridine was photolyzed in the reactor (Fig.2). The concentration of pyridine was measured by HPLC and the decay dynamics was calculated.*

$$I_\lambda = \frac{k'[PNA]_0 l}{1000\Phi(1-10^{-\varepsilon_\lambda l[PNA]_0})} \tag{4}$$

$$\Phi = 0.29[pyr] + 0.00029 \tag{5}$$

*in equation (4) and (5),*

*k'-First order reaction rate constant of pyr ($s^{-1}$);*

*$[PNA]_0$-Initial molar concentration of pyr;*

*$\varepsilon_\lambda$- molar absorbance index of PNA at the wavelength of $\lambda$ ($M^{-1} \cdot cm^{-1}$);*

*l-Optical path (1 cm);*

*$\Phi$-Quantum yield of PNA (mol $\cdot$ einstein$^{-1}$).*" in Text S2 in the improved paper.

➢ *Laszakovits, J.R., Berg, S.M., Anderson, B.G., O'Brien, J.E., Wammer, K.H., Sharpless, C.M.: p-Nitroanisole/Pyridine and p-Nitroacetophenone/Pyridine Actinometers Revisited: Quantum Yield in Comparison to Ferrioxalate, Environ. Sci. Tech. Let., 4, 11-14, http://dx.doi.org/ 10.1021/acs.estlett.6b00422, 2017.*

*2. Line 128: what was the dilution factor?*

The dilution factor is the dilution ratio of extracts and the specific dilution factor has been added in Table S2.

**Table S2.** *OC concentration of samples for optical analysis.*

| | POA | | | | Ambient PM | | |
|---|---|---|---|---|---|---|---|
| Sample ID | $c_{WSOC}$/ppm | $c_{WISOC}$/ppm | Dilution factor (WSOC/WISOC) | Sample ID | $c_{WSOC}$/ppm | $c_{WISOC}$/ppm | Dilution factor (WSOC/WISOC) |
| 1-0h | 2.99 | 6.86 | 40/40 | 9-0h | 59.12 | 7.42 | 1/5 |
| 1-2h | 2.91 | 2.87 | 40/40 | 9-2h | 65.85 | 5.48 | 1/5 |
| 1-6h | 3.13 | 2.52 | 40/40 | 9-6h | 67.39 | 8.30 | 1/5 |
| 1-12h | 3.51 | 2.67 | 40/40 | 9-12h | 55.90 | 6.36 | 1/5 |
| 1-24h | 3.76 | 2.68 | 40/40 | 9-24h | 54.41 | 7.20 | 1/5 |
| 1-3d | 3.02 | 2.15 | 40/40 | 9-3d | 76.46 | 4.41 | 1/1 |
| 1-7d | 3.00 | 2.24 | 40/40 | 9-7d | 63.24 | 2.95 | 1/1 |
| 2-0h | 3.60 | 2.96 | 40/40 | 10-0h | 52.97 | 3.52 | 1/5 |
| 2-2h | 4.19 | 2.85 | 40/40 | 10-2h | 54.92 | 7.25 | 1/5 |
| 2-6h | 4.00 | 3.22 | 40/40 | 10-6h | 61.99 | 7.12 | 1/5 |
| 2-12h | 3.60 | 0.73 | 40/40 | 10-12h | 53.09 | 4.94 | 1/5 |
| 2-24h | 4.13 | 3.78 | 40/40 | 10-24h | 48.29 | 4.68 | 1/5 |
| 2-3d | 3.38 | 4.46 | 40/40 | 10-3d | 48.15 | 4.01 | 1/1 |
| 2-7d | 3.31 | 2.64 | 40/40 | 10-7d | 53.70 | 3.82 | 1/1 |
| 3-0h | 5.86 | 3.15 | 40/60 | 11-0h | 6.62 | 2.79 | 10/10 |
| 3-2h | 6.13 | 3.31 | 40/60 | 11-2h | 4.99 | 4.61 | 10/10 |
| 3-6h | 6.31 | 5.26 | 40/40 | 11-6h | 4.10 | 4.22 | 10/10 |
| 3-12h | 6.20 | 2.51 | 40/40 | 11-12h | 5.01 | 1.67 | 10/10 |
| 3-24h | 5.19 | 5.44 | 40/40 | 11-24h | 5.59 | 2.84 | 10/10 |
| 3-3d | 5.02 | 4.09 | 40/40 | 11-3d | 3.66 | 0.47 | 1/10 |
| 3-7d | 4.70 | 4.90 | 40/40 | 11-7d | 5.34 | 1.08 | 1/10 |
| 4-0h | 4.22 | 2.40 | 40/40 | 12-0h | 5.65 | 3.62 | 10/10 |
| 4-2h | 4.40 | 2.90 | 40/40 | 12-2h | 3.75 | 3.81 | 10/10 |
| 4-6h | 4.02 | 2.90 | 40/40 | 12-6h | 7.15 | 4.38 | 10/10 |
| 4-12h | 3.15 | 5.38 | 40/40 | 12-12h | 4.98 | 3.17 | 10/10 |
| 4-24h | 3.94 | 2.62 | 40/40 | 12-24h | 4.54 | 2.92 | 10/10 |
| 4-3d | 3.22 | 2.98 | 40/40 | 12-3d | 4.03 | 1.42 | 1/1 |
| 4-7d | 3.29 | 2.30 | 40/40 | 12-7d | 5.84 | 1.45 | 10/10 |
| 5-0h | 5.02 | 2.24 | 40/60 | 13-0h | 59.68 | 2.90 | 1/5 |
| 5-2h | 4.74 | 3.82 | 40/40 | 13-2h | 57.95 | 3.71 | 1/5 |
| 5-6h | 5.26 | 4.02 | 40/40 | 13-6h | 50.79 | 5.19 | 1/5 |
| 5-12h | 5.46 | 3.94 | 40/40 | 13-12h | 52.57 | 3.24 | 1/5 |
| 5-24h | 5.21 | 4.13 | 40/40 | 13-24h | 54.15 | 1.34 | 1/5 |
| 5-3d | 4.72 | 4.85 | 40/40 | 13-3d | 55.65 | 1.85 | 1/5 |
| 5-7d | 3.67 | 3.53 | 40/40 | 13-7d | 55.99 | 1.74 | 1/1 |
| 6-0h | 5.23 | 2.46 | 40/60 | 14-0h | 58.12 | 1.86 | 1/5 |
| 6-2h | 5.52 | 2.58 | 40/60 | 14-2h | 47.04 | 4.06 | 1/5 |
| 6-6h | 4.49 | 4.50 | 40/40 | 14-6h | 48.95 | 2.13 | 1/5 |
| 6-12h | 4.28 | 3.72 | 40/40 | 14-12h | 49.47 | 2.80 | 1/5 |
| 6-24h | 4.23 | 3.76 | 40/40 | 14-24h | 39.93 | 3.70 | 1/5 |
| 6-3d | 4.01 | 4.14 | 40/40 | 14-3d | 29.66 | 0.63 | 1/1 |
| 6-7d | 3.64 | 3.42 | 40/40 | 14-7d | 49.12 | 0.59 | 1/1 |
| 7-0h | 9.66 | 3.59 | 40/80 | 15-0h | 42.22 | 2.67 | 1/40 |
| 7-2h | 6.75 | 3.88 | 40/80 | 15-2h | 32.35 | --- | 1/40 |
| 7-6h | 8.24 | 4.23 | 40/80 | 15-6h | 26.49 | 4.96 | 1/40 |

| | POA | | | | Ambient PM | | |
|---|---|---|---|---|---|---|---|
| Sample ID | $c_{WSOC}$/ppm | $c_{WISOC}$/ppm | Dilution factor (WSOC/WISOC) | Sample ID | $c_{WSOC}$/ppm | $c_{WISOC}$/ppm | Dilution factor (WSOC/WISOC) |
| 7-12h | 9.33 | 4.26 | 40/80 | 15-12h | 32.87 | 1.62 | 1/40 |
| 7-24h | 7.62 | 4.06 | 40/80 | 15-24h | 26.60 | 2.60 | 1/40 |
| 7-3d | 7.23 | 3.70 | 40/80 | 15-3d | 26.04 | --- | 1/1 |
| 7-7d | 6.37 | 5.21 | 40/80 | 15-7d | 33.67 | 3.06 | 1/1 |
| 8-0h | 9.43 | 4.02 | 40/80 | 16-0h | 38.66 | 0.93 | 1/10 |
| 8-2h | 9.34 | 2.24 | 40/160 | 162h | 27.01 | 3.07 | 1/10 |
| 8-6h | 9.34 | 4.47 | 40/80 | 16-6h | 29.22 | 2.56 | 1/20 |
| 8-12h | 9.13 | 4.30 | 40/80 | 16-12h | 31.84 | 1.24 | 1/10 |
| 8-24h | 8.51 | 4.41 | 40/80 | 16-24h | 44.70 | 1.19 | 1/10 |
| 8-3d | 6.90 | 5.54 | 40/80 | 16-3d | 27.32 | 0.27 | 1/1 |
| 8-7d | 7.09 | 4.83 | 40/80 | 16-7d | 50.43 | 1.05 | 1/1 |

110

*3. Line 134: How many samples were used to create the model? Were water soluble and methanolic samples combined in the dataset to create the PARAFC model? How were the EEMs of the methanolic extract measured?*

111 samples were used to create the model. In order to compare the change characteristics of water-

115 soluble and water-insoluble chromophores during photolysis, water soluble and methanolic samples combined in the dataset to create the PARAFC model. The methanol background was analyzed and the background was subtracted. The analytical method of methanolic extract is "*The absorption spectra were recorded in the wavelength range of 200-600 nm. The range of excitation wavelength was 200-600 nm and the range of excitation wavelength was 250-800 nm. The interval was 5 nm.*

120 *The exposure time was 0.5 s*" in section 2.5.

We have added "*111 samples were used to create the model. WSOM and WISOM were combined in the dataset to create the PARAFC model*" in section 2.5 in the improved paper.

We have corrected "*The background samples were also analyzed using the same method and the background signals were subtracted from the sample signals*" to "*Water and methanol background*

125 *samples were analyzed using the same method and the background signals were subtracted from the sample signals*" in section 2.5 in the improved paper.

*4. Line 150: Was is isocratic and at what retention time did TMP elute?*

Based on previous studies (Canonica et al., 2000; Kaur & Anastasio, 2018), we optimized the

130 previously reported method. The elution process is isocratic. The retention time is 14.5 min.

We have added "*The retention time is 14.5 min*" in section 2.6 in the improved paper.

➢ Canonica, S., Hellrung, B., Wirz, J.: Oxidation of Phenols by Triplet Aromatic Ketones in Aqueous Solution, J. Phys. Chem. A, 104, 1226-1232, http://dx.doi.org/10.1021/jp9930550, 2000.

135 ➢ Kaur, R., Anastasio, C.: First Measurements of Organic Triplet Excited States in Atmospheric Waters. Environ. Sci. Tech., 52, 5218-5226, http://dx.doi.org/10.1021/acs.est.7b06699, 2018.

*5. Figure 3: The raw absorbance should be converted to apparent absorption coefficient a, so that it is normalized to pathlength.*

140 Fluorescence spectra of COM was measured, but we could not obtain the molecular composition. Therefore, it is difficult to obtain the apparent absorption coefficient. However, the absorbance has been normalized. We have corrected the **Fig.3** in improved paper.

[Figure]

145 *6. Line 220: It is very easy to contaminate fluorescence samples that than show protein like fluorophores. How did the blank samples compare? Was this protein-like fluorescence apparent in controls? This needs to be carefully addressed so that QA/QC can be assessed.*

As shown in the Fig.R1, the first figure is the background. The second figure is the extract of POA. The background contributes essentially nothing to the fluorescence intensity of sample at the peaks
150 of Ex./Em. = 230/300 nm. Therefore, the experimental results are reliable.

[Figure]

Fig. R1. Comparison of the EEMs of background and sample. The first figure is the background. The second figure is the extract of POA.

 *7. Line 240: Triplet state of what?*

We have corrected "*The photochemical activity is characterized by triplet state and singlet oxygen*" to "*The photochemical activity is characterized by the generating ability of $^3COM*$ and singlet oxygen*" in section 3.3 in improved paper.

 *8. Line 270-280: It is well known that singlet oxygen is photochemically generated and very well correlated to UV-vis absorption, so this all seems to me a generic trend that has been described low-energy 3COM* even exists and less so being the main precursor of singlet oxygen, but of course this is debatable.*

As mentioned by the reviewer, singlet oxygen is photochemically generated. Previous study has
 suggested the $^3COM*$ induce $^1O_2$ significantly (Schematic R1).

We obtained the mechanism and characteristic of $^3COM*$ inducing $^1O_2$ by the method of combining capturing agent and EPR. There are obviously different of $^1O_2$ in the POA and ambient PM after $^3COM*$ was quenched by sorbic acid. Therefore, we propose that the generating ability of $^1O_2$ was affected by $^3COM*$ energy in POA and ambient PM.

$$^3CDOM^* + O_2 \xrightarrow{k^q_{^3CDOM^*,O_2}} \begin{cases} \xrightarrow{k^{SO}_{^3CDOM^*,O_2}} CDOM + {}^1O_2 & \text{singlet oxygen formation} \\ \xrightarrow{k^{EL}_{^3CDOM^*,O_2}} CDOM + O_2 & \text{excitation energy loss} \\ \xrightarrow{k^{RQ}_{^3CDOM^*,O_2}} Products & \text{reactive quenching} \end{cases}$$

Schematic R1. Deactivation Pathways for $^3COM*$ in the Presence of Oxygen (Rosario-Ortiz et al., 2016)

➢ Rosario-Ortiz, F. L., and Canonica, S.: Probe Compounds to Assess the Photochemical Activity of Dissolved Organic Matter, Environ. Sci. Technol., 50, 12532-12547, http://dx.doi.org/10.1021/acs.est.6b02776, 2016.

**Response to Anonymous Referee #2**

*The manuscript 'Photodegradation of atmospheric chromophores: type conversion and changes in photochemical reactivity' provides results on the photochemical aging of atmospheric aerosols. The results include OC/EC analysis, parallel factor (PARAFAC) analysis of excitation-emission matrices, and photosensitization of singlet oxygen with each measured as a function of solar*

180 *irradiation. The results are of interest and merit eventual publication in the atmospheric chemistry literature, however, the novelty, writing, and presentation require significant improvement and does not meet the standard for publication in Atmospheric Chemistry and Physics in its current state. My comments are outlined below.*

185 We appreciate the comments from reviewer. We appreciate the positive evaluation of this work. According to the reviewer's comments, we have revised this paper. The details are as follows. *The blue italics are comments of reviews. The red italics are improvements and original text of reviews.* The black font are responses.

190 *General comment: The manuscript would benefit greatly from a substantial revision to improve the writing quality (see minor comments for extensive recommended changes), conceptual framework, and referenced literature, to increase the readability of this work.*

In order to increase the readability of this paper, we have revised the description of the experimental method, the structure of sentences, and the Figures in this paper. We have improved our writing.

195 For example,

We have corrected "*A high-purity quartz reactor was designed for the photolysis experiment (Fig.1a). A rubber gasket was embedded on the upper edge of the reactor. The reactor was clamped with a high-purity quartz cover to form a sealed environment. Two vents were designed in the low position of the reactor. The vents were connected to water circulator to ensure that the temperature*

200 *was about 25°C in the reactor*" to "*The material of reactor is quartz (Fig.2a) and the reactor was designed for photolysis experiment. The reactor was sealed through clamping a quartz cover to reactor. Two air vents were used to air exchange and Two water cycle vents were connected to water circulator to ensure that the temperature was about 25°C in the reactor*" in section 2.2 in the improved paper.

205 We have corrected "*Wheat straw, corn straw, rice straw and wood were burned at about 500 ℃ in the tube stove*" to "*As shown in Fig.1, Wheat straw, corn straw, rice straw and wood were burned in the annular combustion chamber when temperatures rose to 500 ℃*" in section 2.1 in the improved paper.

We have corrected "*The low attenuation result from COM have undergone a long-term atmospheric*

210 *aging process and the water-soluble COM are easier to photolysis*" to "*The results suggest that COM have undergone a long-term atmospheric aging and water-soluble COM have greater ability to be photolyzed*" in section 3.2 in the improved paper.

We have added the Fig.1 in the improved paper.

[Figure]

215

We have corrected the Fig.5 in the improved paper.

[Figure]

220 *Major comments: 1) The title of this manuscript could be revised to remove the 'type conversion' terminology which is ambiguous and not commonly used. I would recommend making it more explicit that you are talking about oxidation by revising to: Photodegradation of Atmospheric Chromophores: Changes in Oxidation State and Photochemical Reactivity.*

We have corrected "*Photo-degradation of atmospheric chromophores: type conversion and*
225 *changes in photochemical reactivity*" to "*Photodegradation of Atmospheric Chromophores: Changes in Oxidation State and Photochemical Reactivity*".

*2) In the first paragraph of section 3.2, the results on the decrease in absorbance vs. fluorescence (or TFV) are not well separated and it is extremely difficult to separate when the authors are*
230 *referring to absorbance or the fluorescence of these. It appears decay constants are provided for TFV but nor for absorbance. . .. Do the photolysis decay kinetics differ significantly for the chromophores vs. fluorophores? Or does the absorbance decay at the same rate as TFV?*

We have corrected Fig.3. The decay of absorbance is different from TFV. We added the figure and data of absorbance at 365 nm to separate the attenuation characteristics of absorbance.

235 We have added "*As shown in the scatter plot (**Fig.4**), absorbance decreases significantly during photolysis. The decay kinetics of absorbance is different to fluorophores. The attenuation trend is inconstant, so the decay kinetics do not be mathematical analyzed and the absorbance also could confirm the photo-degradation of COM*" in section 3.2 in the improved paper.

[Figure]

240

*3) What are the estimated atmospheric photolysis lifetimes including the factor of 1.2- 1.3 mentioned in Sec. 2.2? Also according to Figure S1 in the supplement, your light source has almost no flux*

*from 300-350 nm where the aerosols sample absorb most strongly and differs substantially from the solar spectrum. This should be addressed at some point in the manuscript.*

245 The light intensity is various in different areas. The light intensity used in the experiment is approximately equal to the intensity in Xi'an and the actual lifetime can be estimated by multiplying 1.2-1.3. A more accurate assessment should be made in the actual atmospheric environment. We have corrected the Figure S1 in the improved paper.

[Figure]

250

*4) On pages 8-9, the impacts of photolysis on the EEMs and individual components could be discussed in much more detail with comparison to more literature. Here are some additional references on this topic:*

These references are very valuable to our study. Compared with previous study (Murphy et al.,
255 2018), the results suggest that the number or shape of fluorophores do not change during COM photolysis in this paper.

[Figure]

Fig.R2 Fluorescence spectra of POA before and after photolysis. (a) COM before photolysis. (b) COM after photolysis.

[Figure]

260

Fig.R3 Fluorescence spectra of Ambient PM before and after photolysis. (a) COM before photolysis. (b) COM after photolysis.

265 We have added "*As shown in Fig.4, the attenuation of fluorescence is mathematical analyzed and the number or shape of fluorophores do not change during COM photolysis (Murphy et al., 2018)*" in section 3.2 and "*COM could also be generated after light exposure (Harrison et al., 2020)*" in section 1 in improved paper.

270
➢ *Murphy, K. R.; Timko, S. A.; Gonsior, M.; Powers, L. C.; Wünsch, U. J.; Stedmon, C. A. Photochemistry Illuminates Ubiquitous Organic Matter Fluorescence Spectra. Environ. Sci. Technol. 2018, 52 (19), 11243–11250*

➢ *Harrison, A. W.; Waterson, A. M.; De Bruyn, W. J. Spectroscopic and Photochemical Properties of Secondary Brown Carbon from Aqueous Reactions of Methylglyoxal. ACS Earth Space Chem. 2020, 4 (5), 762–773.*

275

*5) In Figure 3A, the absorbances of WSOM and WISOM are nearly identical in the POA samples but differ by a factor of 10 in ambient PM. Is this a genuine difference in optical properties of these fractions i.e. does the water-soluble fraction actually absorb much more strongly than the water-insoluble fraction? This would be in contrast to previous brown carbon literature worth discussing*
280 *in the manuscript. The use of mass absorption efficiency (MAE) is mentioned in the Supplement but not in the manuscript.*

We have corrected Figure 3.

POA samples were generated in laboratory but the ambient PM was collected in actual atmosphere. The ambient PM has undergone a long-term aging. Therefore, light-absorbing substance in WISOM
285 could be photolyzed and transformed so that WISOM has lower absorbances.

As shown in the scatter plot, absorbance decreases significantly during photolysis process, but the attenuation trend is inconstant. Therefore, MAE could be reference index but could not be mention the COM photo-degradation directly.

[Figure]

*6) Are the WSOM and WISOM combined for the PARAFAC analysis? I would expect that there would be some differences in the fluorophores present in each fraction. See this recent ACP paper where there are clear differences in the water-soluble and methanol-soluble fractions:*

In order to illustrate the difference of chromophores, 111 water-soluble and water-insoluble samples were used to create the PARAFAC model. Chen's study (Chen et al., 2020) has illustrated the distribution characteristics of fluorophores in WSOM/WISOM. Tang's study (Tang et al., 2020) has been read and referenced in the improved paper.

We have added "*Tang et al. (2020) study the chromophores in water-soluble and water-insoluble samples, respectively. However, based on the Chen's study (2020), water-soluble and water-insoluble samples were combined to create the PARAFAC model so that illustrate the distribution of chromophores in WSOM and WISOM*" in section 3.2 in the improved paper.

We have added "*111 samples were used to create the model. WSOM and WISOM were combined in the dataset to create the PARAFC model*" in section 2.5 in improved paper.

➢ Chen, Q.C., Li, J.W., Hua, X.Y., Jiang, X.T., Mu, Z., Wang, M.M., Wang, J., Shan, M., Yang, X.D., Fan, X.J., Song, J.Z., Wang, Y.Q., Guan, D.J., Du, L.: Identification of species and sources of atmospheric chromophores by fluorescence excitation-emission matrix with parallel factor analysis, Sci. Total Environ., 718, 10, http://dx.doi.org/10.1016/j.scitotenv.2020.137322, 2020.

➢ Tang, J., Li, J., Su, T., Han, Y., Mo, Y.Z., Jiang, H.X., Cui, M., Jiang, B., Chen, Y.J., Tang, J.H., Song, J.Z., Peng, P.A., Zhang, G.: Molecular compositions and optical properties of

310 dissolved brown carbon in biomass burning, coal combustion, and vehicle emission aerosols illuminated by excitation-emission matrix spectroscopy and Fourier transform ion cyclotron resonance mass spectrometry analysis. Atmos. Chem. Phys. 20, 2513-2532, http://dx.doi.org/10.5194/acp-20-2513-2020, 2020.

315 *7) Tabulation of the relative changes in each of the components in Figure 4B-C would improve the presentation of the results. This is information is all packed into the paragraph starting at Line 222 and is difficult to parse.*

We have added the Table S4 and S5 in the SI.

320 **Table S4.** *Proportion of COM in POA.*

| | | WSOM | | | | WISOM | | | |
|---|---|---|---|---|---|---|---|---|---|
| | | C1 | C2 | C3 | C4 | C1 | C2 | C3 | C4 |
| Wheat Straw | 0 h | 0.44 | 0.02 | 0.45 | 0.10 | 0.15 | 0.42 | 0.42 | 0.01 |
| | 2 h | 0.52 | 0.00 | 0.42 | 0.06 | 0.17 | 0.40 | 0.41 | 0.02 |
| | 6 h | 0.54 | 0.00 | 0.41 | 0.05 | 0.19 | 0.39 | 0.40 | 0.02 |
| | 12 h | 0.56 | 0.00 | 0.40 | 0.04 | 0.27 | 0.20 | 0.47 | 0.07 |
| | 24 h | 0.60 | 0.00 | 0.37 | 0.03 | 0.27 | 0.21 | 0.46 | 0.06 |
| | 3 d | 0.67 | 0.00 | 0.31 | 0.02 | 0.29 | 0.35 | 0.34 | 0.02 |
| | 7 d | 0.75 | 0.01 | 0.24 | 0.01 | 0.36 | 0.35 | 0.28 | 0.02 |
| Corn Straw | 0 h | 0.47 | 0.11 | 0.33 | 0.10 | 0.30 | 0.35 | 0.33 | 0.03 |
| | 2 h | 0.51 | 0.10 | 0.32 | 0.07 | 0.21 | 0.37 | 0.38 | 0.03 |
| | 6 h | 0.54 | 0.08 | 0.32 | 0.06 | 0.23 | 0.35 | 0.38 | 0.04 |
| | 12 h | 0.57 | 0.08 | 0.30 | 0.05 | 0.25 | 0.34 | 0.38 | 0.04 |
| | 24 h | 0.62 | 0.07 | 0.28 | 0.03 | 0.26 | 0.36 | 0.35 | 0.03 |
| | 3 d | 0.68 | 0.03 | 0.28 | 0.01 | 0.30 | 0.34 | 0.34 | 0.03 |
| | 7 d | 0.72 | 0.01 | 0.26 | 0.01 | 0.35 | 0.32 | 0.31 | 0.02 |
| Rice Straw | 0 h | 0.42 | 0.08 | 0.39 | 0.11 | 0.15 | 0.42 | 0.42 | 0.02 |
| | 2 h | 0.52 | 0.03 | 0.38 | 0.08 | 0.16 | 0.41 | 0.41 | 0.02 |
| | 6 h | 0.52 | 0.03 | 0.37 | 0.08 | 0.19 | 0.40 | 0.40 | 0.02 |
| | 12 h | 0.57 | 0.03 | 0.34 | 0.06 | 0.19 | 0.40 | 0.39 | 0.02 |
| | 24 h | 0.57 | 0.02 | 0.35 | 0.06 | 0.20 | 0.39 | 0.39 | 0.02 |
| | 3 d | 0.63 | 0.00 | 0.32 | 0.04 | 0.26 | 0.35 | 0.37 | 0.02 |
| | 7 d | 0.70 | 0.00 | 0.26 | 0.03 | 0.33 | 0.33 | 0.32 | 0.01 |
| Wood | 0 h | 0.30 | 0.06 | 0.51 | 0.13 | 0.12 | 0.40 | 0.48 | 0.01 |
| | 2 h | 0.34 | 0.02 | 0.53 | 0.12 | 0.12 | 0.41 | 0.47 | 0.00 |
| | 6 h | 0.36 | 0.01 | 0.51 | 0.12 | 0.13 | 0.40 | 0.47 | 0.00 |
| | 12 h | 0.39 | 0.00 | 0.50 | 0.11 | 0.13 | 0.40 | 0.46 | 0.00 |
| | 24 h | 0.40 | 0.01 | 0.49 | 0.10 | 0.13 | 0.42 | 0.46 | 0.00 |
| | 3 d | 0.40 | 0.00 | 0.48 | 0.12 | 0.15 | 0.37 | 0.47 | 0.01 |
| | 7 d | 0.49 | 0.00 | 0.41 | 0.10 | 0.15 | 0.36 | 0.48 | 0.00 |

| | | WSOM | | | | WISOM | | | |
|---|---|---|---|---|---|---|---|---|---|
| | | C1 | C2 | C3 | C4 | C1 | C2 | C3 | C4 |
| Wheat Straw | 0 h | 0.01 | 0.86 | 0.05 | 0.08 | 0.26 | 0.42 | 0.32 | 0.00 |
| | 2 h | 0.00 | 0.86 | 0.05 | 0.09 | 0.31 | 0.40 | 0.27 | 0.01 |
| | 6 h | 0.00 | 0.88 | 0.04 | 0.09 | 0.30 | 0.41 | 0.28 | 0.01 |
| | 12 h | 0.01 | 0.87 | 0.03 | 0.08 | 0.31 | 0.40 | 0.27 | 0.01 |
| | 24 h | 0.03 | 0.86 | 0.03 | 0.08 | 0.30 | 0.41 | 0.28 | 0.01 |
| | 3 d | 0.08 | 0.83 | 0.02 | 0.07 | 0.33 | 0.39 | 0.26 | 0.02 |
| | 7 d | 0.10 | 0.84 | 0.00 | 0.06 | 0.34 | 0.33 | 0.29 | 0.04 |
| Corn Straw | 0 h | 0.34 | 0.49 | 0.15 | 0.02 | 0.31 | 0.44 | 0.24 | 0.01 |
| | 2 h | 0.59 | 0.11 | 0.30 | 0.00 | 0.26 | 0.48 | 0.26 | 0.00 |
| | 6 h | 0.59 | 0.14 | 0.27 | 0.00 | 0.29 | 0.47 | 0.24 | 0.00 |
| | 12 h | 0.61 | 0.14 | 0.25 | 0.00 | 0.31 | 0.47 | 0.21 | 0.00 |
| | 24 h | 0.63 | 0.14 | 0.23 | 0.00 | 0.33 | 0.48 | 0.19 | 0.00 |
| | 3 d | 0.36 | 0.51 | 0.11 | 0.02 | 0.36 | 0.47 | 0.17 | 0.01 |
| | 7 d | 0.08 | 0.89 | 0.00 | 0.04 | 0.39 | 0.44 | 0.16 | 0.01 |
| Rice Straw | 0 h | 0.19 | 0.71 | 0.06 | 0.04 | 0.32 | 0.48 | 0.20 | 0.00 |
| | 2 h | 0.26 | 0.63 | 0.07 | 0.04 | 0.35 | 0.44 | 0.20 | 0.01 |
| | 6 h | 0.31 | 0.59 | 0.07 | 0.03 | 0.37 | 0.44 | 0.18 | 0.01 |
| | 12 h | 0.30 | 0.55 | 0.06 | 0.09 | 0.39 | 0.40 | 0.19 | 0.03 |
| | 24 h | 0.37 | 0.56 | 0.05 | 0.03 | 0.41 | 0.42 | 0.16 | 0.01 |
| | 3 d | 0.52 | 0.41 | 0.06 | 0.01 | 0.46 | 0.34 | 0.18 | 0.02 |
| | 7 d | 0.63 | 0.31 | 0.06 | 0.01 | 0.51 | 0.11 | 0.27 | 0.11 |
| Wood | 0 h | 0.08 | 0.80 | 0.07 | 0.06 | 0.25 | 0.46 | 0.30 | 0.00 |
| | 2 h | 0.12 | 0.74 | 0.08 | 0.06 | 0.28 | 0.42 | 0.29 | 0.01 |
| | 6 h | 0.13 | 0.74 | 0.07 | 0.06 | 0.21 | 0.40 | 0.38 | 0.02 |
| | 12 h | 0.20 | 0.68 | 0.07 | 0.04 | 0.32 | 0.41 | 0.26 | 0.01 |
| | 24 h | 0.27 | 0.62 | 0.07 | 0.03 | 0.35 | 0.42 | 0.23 | 0.00 |
| | 3 d | 0.46 | 0.44 | 0.09 | 0.02 | 0.40 | 0.39 | 0.20 | 0.01 |
| | 7 d | 0.62 | 0.27 | 0.11 | 0.00 | 0.40 | 0.32 | 0.25 | 0.04 |

325

*8) The paper demonstrates that low energy triplets are the main precursor for singlet oxygen , but do very high energy triplets form singlet oxygen with less efficiency in general? It would seem intuitive to me that when the energy gap between singlet and triplet COM is large, the photosensitization reaction will be less efficient.*

As stated earlier, singlet oxygen is photochemically generated. Previous study has suggested the $^3COM^*$ induce $^1O_2$ significantly (Schematic R1). We studied the mechanism and characteristic of $^3COM^*$ inducing $^1O_2$ by the method of combining capturing agent and EPR. There are obviously different in $^1O_2$ in the POA and ambient PM after $^3COM^*$ was quenched by sorbic acid. In POA, high energy $^3COM^*$ was quenched and the $^1O_2$ still exist. However, high energy $^3COM^*$ was quenched but the $^1O_2$ disappears in ambient PM. Therefore, we illustrated that low energy $^3COM^*$ are the main precursor for singlet oxygen. The large energy gap between $^1O_2$ and $^3COM^*$ could be conducive to energy transfer from $^3COM^*$ to $O_2$ and it need to be studied furtherly.

$$^3CDOM^* + O_2 \xrightarrow{k^q_{^3CDOM^*,O_2}} \begin{cases} \xrightarrow{k^{SO}_{^3CDOM^*,O_2}} & CDOM + {}^1O_2 & \text{singlet oxygen formation} \\ \xrightarrow{k^{EL}_{^3CDOM^*,O_2}} & CDOM + O_2 & \text{excitation energy loss} \\ \xrightarrow{k^{RQ}_{^3CDOM^*,O_2}} & Products & \text{reactive quenching} \end{cases}$$

Schematic R1. Deactivation Pathways for $^3COM^*$ in the Presence of Oxygen (Rosario-Ortiz et al., 2016)

➢ Rosario-Ortiz, F. L., and Canonica, S.: Probe Compounds to Assess the Photochemical Activity of Dissolved Organic Matter, Environ. Sci. Technol., 50, 12532-12547, http://dx.doi.org/10.1021/acs.est.6b02776, 2016.

*9) Implication section is short and lacks significance or contextualization in the large amount of literature on the photochemical oxidation of chromophoric organic matter in atmospheric aerosols.*

We think that COM photolysis in aerosol and the effect of COM photolysis on aerosol aging is the most significant implication. However, these two points do not be emphasized in the section 4. The details are as follows.

"*We made a comprehensive study of COM photo-degradation, changes in optical properties and chemical compositions, COM photo-degradation affecting photochemical activity, and COM photo-degradation acting aerosol aging. The properties of COM photo-degradation were revealed. COM photo-degradation could be explained by reduction of carbonaceous components, decrease of light absorption capacity, and attenuation of fluorescence intensity. There are great differences in various COM in aerosols. Therefore, we suggested that the properties COM photo-degradation could be comprehensively characterized by carbonaceous components and optical characteristics.*

*We studied that the photo-degradation could lead to COM decompose and change in types. High-molecular-weight DOM could be decomposed into low-molecular-weight DOM during photolysis. The conversion process of low-oxidation HULIS to high-oxidation HULIS is observed in ambient PM, which reflects the significant influence of photo-degradation on chemical composition. In turn, the attenuation and type conversion of COM provide an important basis to trace the aerosol aging process. Optical properties were also affected by COM photo-degradation.*

*The effects of COM photo-degradation on the photochemical activity in aerosols are studied. We evaluated the effect of COM photo-degradation on the photochemical activity and the ability of triplet state generation and 1O2 yield was chosen to characterize the photochemical activity. Triplet*

365 *state generation ability remain unchanged in ambient PM and increased in POA during aerosol aging. On the one hand, only a small amount of chromophore could generate 3COM\* in aerosols. Thus, COM photo-degradation could not properly illustrate changes in 3COM\*. On the other hand, the energy of capturing agents was closely related to measured 3COM\* and TMP may capture short-lived triplet state. Therefore, chromophores, that could form a short-lived triplet state, may not be reduced or even generated during photolysis. Photo-degradation has a significant*

370 *attenuating effect on the 1O2 yield. Therefore, photolysis and/or conversion of COM could be considered to be the main influence factor for photochemical reaction capacity. COM Photo-degradation indirectly affected the aerosols aging due to the changes in inducing reactive oxygen species. In addition, the photochemical reaction mechanisms and aerosol aging processes are relatively different in aerosols. It may be more useful to distinguish the types of 3COM\* into high*

375 *and low energies, so that the mechanism of COM photochemical reaction can be elucidated. In summary, the aerosol aging process has a remarkable impact on atmospheric photochemistry. Aerosol aging can not only change the type and content of COM, but also change their photochemical activity, which furtherly has a potential impact on the aerosol fate. Different types of aerosols have different aging mechanisms, so the environmental impacts caused by COM should*

380 *also be different*".

*Minor comments:*
*Chromophoric Organic Matters should be 'matter' instead of 'matters'. Consider how making this change will impact verb conjugation*

385 We have corrected "*matters*" to "*matter*" in improved paper.
For example, we have corrected "*excited COM react with organic matters and promote secondary organic aerosols*" to "*excited COM react with organic matter and generate secondary organic aerosols*" in section 1 in the improved paper.
we have corrected "*Organic matters can be decomposed and transformed in aerosol due to*

390 *illumination*" to "*Organic matter can be decomposed and transformed in aerosol due to illumination*" in section 3.1 in the improved paper.

*Line 9: Change 'Furtherly' to 'Furthermore'*
We have corrected "*Furtherly*" to "*Furthermore*" in the improved paper.

395

*Line 12: Change 'particle' to 'particulate'*
We have corrected "*particle*" to "*particulate*" in the improved paper.

*Line 15: Change 'result also enunciate' to 'results also highlight'*
400 We have corrected "*enunciate*" to "*highlight*" in the improved paper.

*Line 30: Change 'improve' to 'is necessary for'*
We have corrected "*improve*" to "*is necessary for*" in the improved paper.

405 *Line 38: Change 'Photochemistry have' to 'Photochemistry has'*

We have corrected "*Photochemistry have*" to "*Photochemistry has*" in the improved paper.

We have corrected "*time*" to "*times*" in the improved paper.

410

We have corrected "*analyzed*" to "*measured*" in the improved paper.

415    We have corrected "*excitation*" to "*emission*" in the improved paper.

We have corrected "*chromophores*" to "*fluorophores*" in the improved paper.

420
We have corrected "*partially photolysis*" to "*undergo partial photolysis*" in the improved paper.

425    We have corrected "*result*" to "*results*" in the improved paper.

We have corrected "*Contrast with*" to "*In contrast to*" in the improved paper.

430
We have corrected "*Ambint PM*" to "*Ambient PM*" in **Fig.2** in the improved paper.

[Figure]

435
We have corrected "*represent an obvious decreasing trend due to aerosol photolysis*" to "*significantly decrease during aerosol photolysis*" in the improved paper.

440    We have corrected "*subduction*" to "*decay*" in the improved paper.

*Line 210: Change 'photolysis' to 'photolyze'*

We have corrected "*photolysis*" to "*photolyze*" in the improved paper.

445 *Line 208-210: Sentence that begins with 'The low attenuation result from COM' is unclear and needs revision*

We have corrected "*The low attenuation result from COM have undergone a long-term atmospheric aging process and the water-soluble COM are easier to photolysis*" to "*The results suggest that COM have undergone a long-term atmospheric aging and water-soluble COM have greater ability*
450 *to be photolyzed*" in the improved paper.

*Line 219: Change 'chromophores' to 'fluorophores'*

We have corrected "*chromophores*" to "*fluorophores*" in the improved paper.

455 *Line 220: Change 'identified as' to 'associated with'*

We have corrected "*identified as*" to "*associated with*" in the improved paper.

*Line 243: Change 'show' to 'shows'*

We have corrected "*show*" to "*shows*" in the improved paper.

460
*Line 243: Change 'do not significant affect' to 'does not significantly affect'*

We have corrected "*do not significant affect*" to "*does not significantly affect*" in the improved paper.

*Line 260: Change 'through the approach of' to 'using'*

465 We have corrected "*through the approach*" to "*using*" in the improved paper.

*Line 261: Change 'EPRs' to 'EPR'*

We have corrected "*EPRs*" to "*EPR*" in the improved paper.

470 *Line 267: Change 'increase by 3 times' to 'increases by a factor of 3'*

We have corrected "*increase by 3 times*" to "*increases by a factor of 3*" in the improved paper.

*Line 277: Line 176: Change 'do' to 'does'*

We have corrected "*do*" to "*does*" in the improved paper.

475
*Line 278: Change 'The mechanism is same as' to 'The mechanism is the same as the'*

We have corrected "*The mechanism is same as*" to "*The mechanism is the same as the*" in the improved paper.

480 *Line 285: Change 'disappear' to 'disappears'*

We have corrected "*disappear*" to "*disappears*" in the improved paper.

*Line 288-289: Consider changing 'restraining' to 'attenuating' or 'inhibiting'. 'Restraining' is an odd word to use for this.*

485    We have corrected "*restraining*" to "*attenuating*" in the improved paper.

*Line 291: Change 'prove' to 'show'*
We have corrected "*prove*" to "*show*" in the improved paper.

490    *Line 293: Change 'prove' to 'study'*
We have corrected "*prove*" to "*study*" in the improved paper.

*Line 305: Change 'reflect' to 'reflects'*
We have corrected "*reflect*" to "*reflects*" in the improved paper.

495

---

## Referee Report (RR1)

The manuscript 'Photodegradation of Atmospheric Chromophores: Changes in Oxidation State and Photochemical Reactivity' provides results on the photochemical aging of atmospheric aerosols (both ambient PM and laboratory generated POA). The results include OC/EC analysis, parallel factor (PARAFAC) analysis of excitation-emission matrices, and photosensitization of $^1O_2$ with each measured as a function of solar irradiation. The manuscript has been improved but still requires refinement in the writing/presentation and explanation. My comments are outlined below.

Major comments:

1) In the first paragraph of section 3.1, the authors go back and forth between water-soluble and water-insoluble organic matter and WSOC and WISOC. Is there a reason the terminology is different? In Line 184-186, I believe the authors mean to say that the WISOC *decomposes more rapidly* in ambient PM than in POA. If so, rephrase accordingly. In addition, the authors say that ambient PM has been subjected sufficient atmospheric oxidation so that OM is not decomposed, however, the WISOC fraction of ambient PM shows significant attenuation after photolysis. These two aspects seem at odds with one another. More explanation of what the attenuation ratio is would be instructive. This doesn't appear in Sec. 2.4 of the Methods or in Sec. 3.1.

2) In Sec. 3.2, Lines 202-206 are very repetitive stating that the absorbance decreases significantly during photolysis in multiple consecutive sentences. Re-write for clarity. Also in this paragraph, the authors state that the absorbance decay is inconstant and cannot be mathematically analyzed. Do the authors mean that the absorbance decrease cannot be fit to a single exponential decay? Can the authors report a total percent decrease in the absorbance at 350 nm instead?

3) As mentioned above the authors claim that the ambient PM samples have been subjected to sufficient atmospheric oxidation (line 196), however in Figure 5C, these samples are dominated by low oxidation HULIS/C2 (especially compared to the POA samples). How do you reconcile these two observations?

4) In the paragraph starting at Line 274, the authors begin using the term 'light excitation' instead of photolysis or illumination which are used in the figures. It would improve the manuscript to make this terminology consistent. Also, in Line 280-281, what is meant by 'POA has certain oxidability'? Re-phrase whatever concept is trying to be conveyed here.

5) The Implications section is still brief and lacks any reference to previous literature. This section could also be improved by broader interpretation of all the included results. For instance, how do the results in Sec. 3.3 on singlet oxygen generation connect with the results on degree of oxidation in Sec. 3.1 and 3.2, i.e. photolysis increases the degree of oxidation in the aerosol samples which in turn leads to a higher capacity for singlet oxygen formation via photosensitization reactions.

Minor Comments:

Line 2: Change 'photosensitiveness' to 'photosensitivity' and 'have' to 'has'

Line 10-13: Re-write this sentence: 'In terms of photochemical reactivity, the triplet state COM *decreases* slightly in ambient particulate matter samples but *increases* in primary organic aerosol (POA) *following photolysis*.

Line 25: Change 'chemistry' to 'chemical'

Line 39-40: Change 'not complete clear' to 'unclear'

Line 58: Change 'participate' to 'participates'

Line 60: Change 'light conditions' to 'solar irradiation'

Line 61: Change 'induce' to 'can generate'

Line 76: Change 'stated' to 'studied'

Line 125: Change 'could refer to the previous literature' to 'has been described previously'

Line 146 Change 'states' to 'state'

Line 229: Change 'study' to 'studied' and 'chromophores' to 'fluorophores' to highlight that you are referring to fluorescence here

Line 230: Change ', respectively' to 'separately'

Line 231: Change 'so that' to 'to'

Line 232: Change 'chromophores' to 'fluorophores'

Line 238: Rewrite as "The composition of the fluorophores changes significantly during the photolysis process."

Line 260: Change 'states' to 'state'

Line 263 Change 'not as expected' to 'unexpected'

Line 274: Change to 'further induce singlet oxygen formation'

Line 278: Change 'is' to 'of'

---

## Referee Report (RR2)

The manuscript 'Photodegradation of Atmospheric Chromophores: Changes in Oxidation State and Photochemical Reactivity' provides results on the photochemical aging of atmospheric aerosols (both ambient PM and laboratory generated POA). The results include OC/EC analysis, parallel factor (PARAFAC) analysis of excitation-emission matrices, and photosensitization of $^1O_2$ with each measured as a function of solar irradiation. The manuscript has been improved sufficiently for publication. I only have minor comments to improve the readability of the manuscript. My comments are outlined below.

**Minor Comments:**

Line 11: Change 'result' to 'results'

Line 15-16: Change 'change the compositions' to 'changes the composition'

Line 25: Change 'originate' to 'originates'

Line 29: Change 'process' to 'processing'

Line 34-35: Change 'chromophores are photo-bleached' to 'chromophore photo-bleaching'

Line 49: Remove 'ability on'

Line 55: Change 'the complex photochemical reaction' to 'complex photochemical reactions'

Line 63-64: 'deactivate quickly with the ways of' to 'deactivates by'

Line 80: Change to 'in *the* laboratory'

Line 132: Change 'Analysis error' to "Error analysis'

Line 182: Change 'opposite' to 'in contrast'

Line 186: Change 'indicate' to 'indicates'

Line 187: Change 'opposite' to 'in contrast'

Line 188-190: This sentence is unclear. Consider re-phrasing for clarity.

Line 217: Change 'consider' to 'considered'

Line 259: Change 'promote' to 'promotes'

Line 296: Change 'lead' to 'leads'

Line 307-308: Change '…chemical compositions, and photochemical activity. The characteristics of COM photo-degradation were revealed.' to '…chemical composition, and photochemical activity to reveal the characteristics of COM photo-degradation.'

Line 320: Change 'dominant' to 'dominate'

Line 335: Change 'it' to 'this'

Line 339" Change 'would be" to 'could'

Line 340: Change 'celebrate' to 'calibrate'. I believe this is the word that is intended.

Line 346: Change 'COM photodegradation have' to "COM photodegradation has a'

---

## Author Response (AR2)

*The introduction cites literature that has nothing to do with atmospheric aerosols and are based on photochemical studies of aquatic chromophoric dissolved organic matter. This is not stated here, and hence the references used here are inappropriate and misleading. Unfortunately the introduction did not improve and does not make correctly use of the primary literature. A lot of photochemical studies on aerosols are still missing and wrong citations have been used. Grammar remains distracting and some of the descriptions are confusing as a result of this. Overall, the manuscript improved slightly but remains poorly written and misused the literature. The overall novelty is still not clear to me, but perhaps the evaluation of the ROS is of merit. I am also concerned to combine optical data that were collected in different solvents because of the known substantial matrix effects associated with different solvent matrices. I still cannot recommend publication of this manuscript in its current state.*

We appreciate the comments from reviewer. According to the reviewer's comments, we have revised this paper. The details are as follows. *The blue italics are comments of reviews. The red italics are improvements and original text of reviews.* The black font are responses.

We have improved the grammar and the descriptions. For example,

(1) We have corrected "*the physical and chemical characteristics of COM change significantly under sunlight exposure*" to "*the optical characteristics and components of COM change significantly under solar irradiation*" in improved paper.

(2) We have corrected "*Sunlight exposure cause the photo-bleaching of COM*" to "*optical properties change significantly due to chromophores are photo-bleached in aerosols*" in improved paper.

(3) We have corrected "*excited COM react with organic matter and generate secondary organic aerosols (Zhao et al., 2015; Saleh et al., 2013; Zhong and Jang, 2014; Lee et al., 2014; Liu et al., 2016)*" to "*For example, COM can be oxidized by hydroxyl radicals (•OH) (Zhao et al., 2015) and the formation of polyols can be attributed to photooxidation of isoprene, which could be initiated by •OH (Claeys et al., 2004)*" in improved paper.

(4) We have corrected "*As shown in the scatter plot (Fig.4), absorbance decreases significantly during photolysis. The decay kinetics of absorbance is different to fluorophores. The attenuation trend is inconstant, so the decay kinetics do not be mathematical analyzed and the absorbance also could confirm the photo-degradation of COM. As shown in Fig.4, the attenuation of fluorescence is mathematical analyzed and the number or shape of fluorophores do not change during COM photolysis*" to "*The attenuations of fluorescence and absorption coefficients are fit to first-order decay. The absorption coefficients decrease by 32.0% and TFV decreases by 71.4% on average. However, as shown in Fig.3, fluorescence intensities increase and decrease in different regions of EEMs (Aiona et al., 2018; Timko et al., 2015)*" in improved paper.

We also deleted the wrong citations. For example,

➢ *Cory, R. M., and McKnight, D. M.: Fluorescence spectroscopy reveals ubiquitous presence of oxidized and reduced quinones in dissolved organic matter, Environ. Sci. Technol., 39, 8142-8149, http://dx.doi.org/10.1021/es0506962, 2005.*

➢ *Del Vecchio, R., and Blough, N. V.: Photobleaching of chromophoricdissolved organic matter in natural waters: kinetics and modeling, Mar. Chem., 78, 231–253, http://dx.doi.org/10.1016/S0304-4203(02)00036-1, 2002.*

➢ *Gonsior, M., Peake, B. M., Cooper, W. T., Podgorski, D., D'Andrilli, J., and Cooper, W. J.: Photochemically induced changes in dissolved organic matter identified by ultrahigh*

> *resolution fourier transform ion cyclotron resonance mass spectrometry, Environ. Sci. Technol., 43, 698-703, http://dx.doi.org/10.1021/es8022804, 2009.*

> *Kieber, R. J., Adams, M. B., Wiley, J. D., Whitehead, R. F., Avery, G. B., Mullaugh, K. M., and Mead, R. N.: Short term temporal variability in the photochemically mediated alteration of chromophoric dissolved organic matter (CDOM) in rainwater, Atmos. Environ., 50, 112-119, http://dx.doi.org/10.1016/j.atmosenv.2011.12.054, 2012.*

> *Vodacek, A., Blough, N. V., DeGrandpre, M. D., Peltzer, E. T., and Nelson, R. K.: Seasonal Variation of CDOM and DOC in the Middle Atlantic Bight: Terrestrial Inputs and Photooxidation, Limnol. Oceanogr., 42, 231-253, http://dx.doi.org/10.1117/12.26643, 1997.*

We also added the citations about aerosol in improved manuscript. For example,

> *Budisulistiorini, S. H.; Riva, M.; Williams, M.; Chen, J.; Itoh, M.; Surratt, J. D.; Kuwata, M.: Light-Absorbing Brown Carbon Aerosol Constituents from Combustion of Indonesian Peat and Biomass. Environ. Sci. Technol., 51, 4415-4423, http://dx.doi.org/10.1021/acs.est.7b00397, 2017.*

> *Atkinson, R.; Baulch, D. L.; Cox, R. A.; Crowley, J. N.; Hampson, R. F.; Hynes, R. G.; Jenkin, M. E.; Rossi, M. J.; Troe, J.: Evaluated kinetic and photochemical data for atmospheric chemistry: Volume II - gas phase reactions of organic species, Atmos. Chem. Phys., 6, 3625-4055, http://dx.doi.org/10.5194/acp-6-3625-2006, 2006.*

> *Carlton, A. G.; Turpin, B. J.; Altieri, K. E.; Seitzinger, S.; Reff, A.; Lim, H. J.; Ervens, B.: Atmospheric oxalic acid and SOA production from glyoxal: Results of aqueous photooxidation experiments, Atmos. Environ., 41, 7588-7602, http://dx.doi.org/10.1016/j.atmosenv.2007.05.035, 2007.*

> *Claeys, M.; Graham, B.; Vas, G.; Wang, W.; Vermeylen, R.; Pashynska, V.; Cafmeyer, J.; Guyon, P.; Andreae, M. O.; Artaxo, P.; Maenhaut, W.: Formation of secondary organic aerosols through photooxidation of isoprene, Science, 303, 1173-1176, http://dx.doi.org/10.1126/science.1092805, 2004.*

> *Mang, S. A.; Henricksen, D. K.; Bateman, A. P.; Andersen, M. P. S.; Blake, D. R.; Nizkorodov, S. A.: Contribution of Carbonyl Photochemistry to Aging of Atmospheric Secondary Organic Aerosol, J. Phys. Chem. A, 112, 8337-8344, http://dx.doi.org/10.1021/jp804376c, 2008.*

The novelty of the paper is (1) highlighting the effect COM photodegradation on carbonaceous components, optical properties, fluorophores in different aerosols; (2) illustrating changes in photochemical activity during COM photodegradation process by the method of probing triplet state and ROS generation; (3) exploring the mechanism of COM photodegradation affecting aerosol aging. In order to clear the overall novelty, we have revised the last paragraph in Sec 1. "*The objectives of the study are (1) to clarify the characteristics of carbonaceous components variation during COM photolysis, (2) to explore the effects of photo-degradation on the components and optical properties of water-soluble and water-insoluble chromophores, and (3) to investigate the effects of COM photo-degradation on photochemical reactivity and aerosol aging (photochemical reactivity is characterized by triplet state and singlet oxygen generation capacity)*".

*Specific comments:*

*1. Line 10: HULIS needs to be defined.*

According to the reviewer's suggestion, we have corrected "*there is a transformation from low oxidation to high oxidation HULIS and high oxidation HULIS*" to "*low oxidation humic-like substance (HULIS) is converted into high oxidation HULIS*" in improved paper.

*2. Line 38: Murphey et al 2018 did not study aerosols. This study was undertaken with dissolved organic matter from different aquatic systems and cannot be directly compared to aerosols.*

According to the reviewer's suggestion, we have deleted the wrong citations.

We have deleted "*Murphy et al. reported that fluorescence intensity of chromophores decreased after 20 h of simulated solar irradiation (Murphy et al., 2018)*".

We have deleted the citations about aquatic systems, for example,

➢ *Cory, R. M., and McKnight, D. M.: Fluorescence spectroscopy reveals ubiquitous presence of oxidized and reduced quinones in dissolved organic matter, Environ. Sci. Technol., 39, 8142-8149, http://dx.doi.org/10.1021/es0506962, 2005.*

➢ *Del Vecchio, R., and Blough, N. V.: Photobleaching of chromophoricdissolved organic matter in natural waters: kinetics and modeling, Mar. Chem., 78, 231–253, http://dx.doi.org/10.1016/S0304-4203(02)00036-1, 2002.*

➢ *Gonsior, M., Peake, B. M., Cooper, W. T., Podgorski, D., D'Andrilli, J., and Cooper, W. J.: Photochemically induced changes in dissolved organic matter identified by ultrahigh resolution fourier transform ion cyclotron resonance mass spectrometry, Environ. Sci. Technol., 43, 698-703, http://dx.doi.org/10.1021/es8022804, 2009.*

➢ *Kieber, R. J., Adams, M. B., Wiley, J. D., Whitehead, R. F., Avery, G. B., Mullaugh, K. M., and Mead, R. N.: Short term temporal variability in the photochemically mediated alteration of chromophoric dissolved organic matter (CDOM) in rainwater, Atmos. Environ., 50, 112-119, http://dx.doi.org/10.1016/j.atmosenv.2011.12.054, 2012.*

➢ *Vodacek, A., Blough, N. V., DeGrandpre, M. D., Peltzer, E. T., and Nelson, R. K.: Seasonal Variation of CDOM and DOC in the Middle Atlantic Bight: Terrestrial Inputs and Photooxidation, Limnol. Oceanogr., 42, 231-253, http://dx.doi.org/10.1117/12.26643, 1997.*

We have added the citations about aerosol, for example,

➢ *Budisulistiorini, S. H.; Riva, M.; Williams, M.; Chen, J.; Itoh, M.; Surratt, J. D.; Kuwata, M.: Light-Absorbing Brown Carbon Aerosol Constituents from Combustion of Indonesian Peat and Biomass. Environ. Sci. Technol., 51, 4415-4423, http://dx.doi.org/10.1021/acs.est.7b00397, 2017.*

➢ *Atkinson, R.; Baulch, D. L.; Cox, R. A.; Crowley, J. N.; Hampson, R. F.; Hynes, R. G.; Jenkin, M. E.; Rossi, M. J.; Troe, J.: Evaluated kinetic and photochemical data for atmospheric chemistry: Volume II - gas phase reactions of organic species, Atmos. Chem. Phys., 6, 3625-4055, http://dx.doi.org/10.5194/acp-6-3625-2006, 2006.*

➢ *Carlton, A. G.; Turpin, B. J.; Altieri, K. E.; Seitzinger, S.; Reff, A.; Lim, H. J.; Ervens, B.: Atmospheric oxalic acid and SOA production from glyoxal: Results of aqueous photooxidation experiments, Atmos. Environ., 41, 7588-7602, http://dx.doi.org/10.1016/j.atmosenv.2007.05.035, 2007.*

➢ *Claeys, M.; Graham, B.; Vas, G.; Wang, W.; Vermeylen, R.; Pashynska, V.; Cafmeyer, J.; Guyon, P.; Andreae, M. O.; Artaxo, P.; Maenhaut, W.: Formation of secondary organic aerosols through photooxidation of isoprene, Science, 303, 1173-1176, http://dx.doi.org/10.1126/science.1092805, 2004.*

➢ *Mang, S. A.; Henricksen, D. K.; Bateman, A. P.; Andersen, M. P. S.; Blake, D. R.; Nizkorodov, S. A.: Contribution of Carbonyl Photochemistry to Aging of Atmospheric Secondary Organic Aerosol, J. Phys. Chem. A, 112, 8337-8344, http://dx.doi.org/10.1021/jp804376c, 2008.*

*3. Line 42: A lot of the cited references are not specific to aerosols and have been used for aquatic CDOM. This is misleading and is a false statement.*

According to the reviewer's suggestion, we have corrected the references. Details are shown in second comment.

*4. Line 62: super oxide and not super-oxygen and hydroxyl radicals and not hydroxyl*

According to the reviewer's suggestion, we have corrected "*hydroxyl (•OH)*" to "*hydroxyl radicals (•OH)*" in improved paper.

For example, we added "*COM can be oxidized by hydroxyl radicals (•OH)*" in improved paper.

*5. Line 91-92: At 500 C, the combustion of organics is supposed to be complete, especially by introducing fresh air, I remain confused why this was done. Also, how do you get POA after achieving complete combustion as it was stated in the text?*

POA samples in the study is not the residue after complete combustion, but the particulate matter generated in the combustion process. Even if the temperature rises to 500 °C, there will be a lot of particulate matter. The combustion process is dynamic and particulate matter generated during the combustion process would be diluted and cooled in the mixing box (Figure S1). As shown in the figure, POA was collected on quartz filters.

[Figure]

*6. Line99-108: The description of the reactor system is highly confusing and even the photos do not help to clarify the setup.*

According to the reviewer's suggestion, we have revised the description of the reactor.

(1) we have corrected "*Two air vents was used to air exchange and Two water cycle vents were connected to water circulator to ensure that the temperature constant in the reactor*" to "*Two air vents were designed in upper side of reactor. Two water cycle vents were designed in lower side and connected with water circulator to keep temperature was about 25°C in the reactor*" in improved Supporting Information.

(2) We have added "*Two capsules (Figure S2(b) & (c) of SI) were designed for triplet state and ROS generation experiment*" in Sec 2.2 in improved paper.

(3) We have corrected the figure and added the figure in improved Supporting Information.

[Figure]

***Figure S2.*** *Schematic diagrams of the photochemical devices. (a) The reactor is used for maintaining the reaction environment. (b) The capsule is used for the experiment of triplet state inducing singlet oxygen. The size of quartz plate is 35×35 mm². The size of the tanks is a radius of 5.6 mm and a depth of 2.5 mm. (c) The capsule is used for triplet state experiments. The reactor is made of quartz. The plugs are made of Teflon. The internal volume is 200 μL.*

The details of reactor could refer to our previous study.

[Figure]

Photochemistry reaction device. The inside diameter of the reactor is approximately 11 cm, and the height is approximately 18 cm. The reactor is connected to a cooling-water circulator to ensure that the water temperature in the reactor was constant and the water temperature is set to 8 °C (Chen et al. 2021).

➢ *Chen, Q.; Mu, Z.; Xu, L.; Wang, M.; Wang, J.; Shan, M.; Fan, X.; Song, J.; Wang, Y.; Lin, P.; Du, L.: Triplet-state organic matter in atmospheric aerosols: Formation characteristics and potential effects on aerosol aging, Atmos. Environ., 252, 118343, https://doi.org/10.1016/j.atmosenv.2021.118343, 2021.*

*7. Line 125-130: How was organic carbon quantified here? It is not clear at all.*

The extracts were analyzed by the OC/EC analyzer.

We have revised section 2.4. Section 2.3 and 2.4 were merged. We have corrected "*The method of organic carbon (OC) analysis could refer to the previous literature. 100 µL extracts were injected on the baked quartz filter (Mu et al., 2019). Then, the wet filters were dried out by a rotary evaporator and the WSOC/WISOC adhered to the filter. Carbonaceous components on the filters were analyzed by the OC/EC online analyzer*" to "*The analytical method of carbonaceous components has been described previously (Mu et al., 2019). Briefly, 100 µL extracts were injected on the baked quartz filter. Then, the wet filters were dried out by a rotary evaporator and the dried filters were analyzed by the OC/EC online analyzer*" in section 2.3 in improved paper.

*8. Line 138: Optical properties are highly dependent on the matrix and it is not expected to be able to directly compare EEMS obtained in methanol versus the one collected in water. It is even more problematic when the dataset is combined to create a PARAFAC model.*

In order to compare the differences of fluorescence components in water-soluble and water-insoluble organic matter, we coupled the water-soluble and water-insoluble components to establish the model. Similar methods have been proved in our previous study (Chen et al., 2021). Chen et al. (2016) compared EEMs of water-soluble and water-insoluble BrC. The result has suggested that solvent had no significant effect on the EEM spectra of complex mixtures in aerosols.

We have stated "*based on the Chen's studies (2020; 2016b), water-soluble and water-insoluble samples were combined to create the PARAFAC model to illustrate the distribution of fluorophores in WSOM and WISOM and solvent had no significant effect on the EEMs of complex mixtures in aerosols*" in Sec 3.2 in improved paper.

➢ *Chen, Q. C.; Li, J. W.; Hua, X. Y.; Jiang, X. T.; Mu, Z.; Wang, M. M.; Wang, J.; Shan, M.; Yang, X. D.; Fan, X. J.; Song, J. Z.; Wang, Y. Q.; Guan, D. J.; Du, L.: Identification of species and sources of atmospheric chromophores by fluorescence excitation-emission matrix with parallel*

*factor analysis, Sci. Total Environ., https://doi.org/10.1016/j.scitotenv.2020.1373222020, 718, 10, 2020.*

➢ *Chen, Q. C.; Ikemori, F.; Mochida, M.: Light Absorption and Excitation–Emission Fluorescence of Urban Organic Aerosol Components and Their Relationship to Chemical Structure. Environ. Sci. Technol., 50, 10859-10868, https://doi.org/10.1021/acs.est.6b02541, 2016.*

*9. Line 146L It still does not state here what triplet state species is referred to.*

Triplet states is important reactive intermediates and it could induce the ROS. Therefore, $^3COM^*$ may affect photochemical process.

We have corrected "*The triplet states generation ability before and after photolysis were studied*" to "*As short-lived reactive intermediates, $^3COM^*$ have an important impact on photochemical process in atmospheric environment (Kaur et al., 2018). Therefore, $^3COM^*$ generation ability before and after photodegradation were studied*" in Sec 2.5 in improved paper.

We have revised "*Aromatic ketones could be excited to generate triplet state ($^3COM^*$) under light conditions (Rosario-Ortiz and Canonica, 2016; Del Vecchio and Blough, 2004; Wenk et al., 2013; Ma et al., 2010). $^3COM^*$ induce reactive oxygen species (ROS), such as singlet oxygen ($^1O_2$), super-oxygen ($\bullet O_2^-$) and hydroxyl ($\bullet OH$), which could drive aerosol aging (Paul Hansard et al., 2010; Szymczak and Waite, 1988; Zhang et al., 2014; Rosario-Ortiz and Canonica, 2016; Sharpless, 2012; Haag and Gassman, 1984)*" to "*Upon light absorption, high-energy singlet state COM ($^1COM^*$) could be generated. $^1COM^*$ deactivate quickly with the ways of emitting photon (fluorescence) and intersystem crossing (triplet state, $^3COM^*$). $^3COM^*$ can generate reactive oxygen species (ROS), such as singlet oxygen ($^1O_2$), super oxide ($\bullet O_2^-$) and $\bullet OH$, which indicate that $^3COM^*$ play a critical role in ROS formation and pollutant attenuation (Paul Hansard et al., 2010; Szymczak & Waite, 1988; Zhang et al., 2014; Rosario-Ortiz and Canonica, 2016; Sharpless, 2012; Haag and Gassman, 1984; Zhou et al., 2019). A lot of DOM, such as aromatic ketones (Canonica et al., 2006; Marciniak et al., 1993), benzophenone (Encinas et al., 1985), and phenanthrene (Wawzonek & Laitinen, 1942), have been identified as the precursor of $^3COM^*$*" in Sec 1 in improved paper.

*10. Figure 4: Absorbance should be normalized to pathlength. Please see Helms, J. R.; Stubbins, A.; Ritchie, J. D.; Minor, E. C.; Kieber, D. J.; Mopper, K., Absorption spectral slopes and slope ratios as indicators of molecular weight, source, and photobleaching of chromophoric dissolved organic matter. Limnology and Oceanography 2008, 53, (3), 955-969. And equation (2) within.*

According to the reviewer's suggestion, we have corrected Fig.4 (Fig.2 in improved paper).

[Figure]

We have added the equation in Supporting Information.

"*Absorption coefficient is calculated as follows (Helms et al., 2008):*

$$a=2.303A/l \quad\quad\quad\quad\quad (4)$$

*In (4), a is absorption coefficient ($m^{-1}$), A is absorbance, and l is path length (m)*".

➢ *Helms, J. R.; Stubbins, A.; Ritchie, J. D.; Minor, E. C.; Kieber, D. J.; Mopper, K.: Absorption spectral slopes and slope ratios as indicators of molecular weight, source, and photobleaching of chromophoric dissolved organic matter, Limnol. Oceanogr., 53, 955-969, https://doi.org/10.4319/lo.2008.53.3.0955, 2008.*

*The manuscript 'Photodegradation of Atmospheric Chromophores: Changes in Oxidation State and Photochemical Reactivity' provides results on the photochemical aging of atmospheric aerosols (both ambient PM and laboratory generated POA). The results include OC/EC analysis, parallel factor (PARAFAC) analysis of excitation-emission matrices, and photosensitization of 1O2 with each measured as a function of solar irradiation. The manuscript has been improved but still requires refinement in the writing/presentation and explanation. My comments are outlined below. Major comments:*

We appreciate the positive comments from reviewer. According to the reviewer's comments, we have revised this paper. The details are as follows. *The blue italics are comments of reviews. The red italics are improvements and original text of reviews.* The black font are responses.

*1) In the first paragraph of section 3.1, the authors go back and forth between water-soluble and water-insoluble organic matter and WSOC and WISOC. Is there a reason the terminology is different? In Line 184-186, I believe the authors mean to say that the WISOC decomposes more rapidly in ambient PM than in POA. If so, rephrase accordingly. In addition, the authors say that ambient PM has been subjected sufficient atmospheric oxidation so that OM is not decomposed, however, the WISOC fraction of ambient PM shows significant attenuation after photolysis. These two aspects seem at odds with one another. More explanation of what the attenuation ratio is would be instructive. This doesn't appear in Sec. 2.4 of the Methods or in Sec. 3.1.*

WSOM and WISOM are defined as extracted organic matter. WSOC and WISOC are defined as carbonaceous component in extracted organic matter.

(1) We have corrected the terminology in Sec 3.1 and we have revised the sentence.

We have corrected "*The results show that both water-soluble and water-insoluble organic matter undergo partial photolysis in POA samples (Fig.3A), with an average decrease of 22.1% and 3.5%, respectively*" to "*In POA, water soluble and water insoluble organic carbon (WSOC and WISOC) decrease by 22.1% and 3.5%, respectively.*" in Sec 3.1 in improved paper.

(2) We have added more explanation in Sec 3.1.

For example, we have added "*there is a process of OC1 translating into pyrolysis carbon (OPC)*" in Sec 3.1 in improved paper.

We have corrected "*The result reflects that different carbonaceous components have the similar abilities of photodegradation in ambient PM. Organic matter with high molecular weight is photocomposed to small molecular weight and the molecular weight tend to be consistent following the photodegradation*" in Sec 3.1 in improved paper.

*2) In Sec. 3.2, Lines 202-206 are very repetitive stating that the absorbance decreases significantly during photolysis in multiple consecutive sentences. Re-write for clarity. Also in this paragraph, the authors state that the absorbance decay is inconstant and cannot be mathematically analyzed. Do the authors mean that the absorbance decrease cannot be fit to a single exponential decay? Can the authors report a total percent decrease in the absorbance at 350 nm instead?*

The attenuations of absorption coefficients are fit to first-order decay and we have revised Fig.4 (Fig.2 in improved paper).

[Figure]

We have corretcd "*Both absorbance and total fluorescence volume (TFV, RU-nm2/m3) significantly decrease during aerosol photolysis (Fig.4). Changes in optical properties are shown in Figure S3, S4 and S5. The decrease of absorbance confirm that COM are photo-bleached (Duarte et al., 2005) and the decay function of photolysis on absorbance is significant (Aiona et al., 2018). As shown in the scatter plot (**Fig.4**), absorbance decreases significantly during photolysis*" to "*As shown in Fig.2, both absorption coefficients and total fluorescence volume (TFV, RU-nm2/m3) significantly decrease following aerosol photodegradation, which suggest that COM are photo-bleached (Aiona et al., 2018; Duarte et al., 2005; Liu et al., 2016). The attenuations of fluorescence and absorption coefficients are fit to first-order decay. The absorption coefficients decrease by 32.0% and TFV decreases by 71.4% on average.*" in Sec 3.2.

*3) As mentioned above the authors claim that the ambient PM samples have been subjected to sufficient atmospheric oxidation (line 196), however in Figure 5C, these samples are dominated by low oxidation HULIS/C2 (especially compared to the POA samples). How do you reconcile these two observations?*

(1) The components of ambient PM are affected by various sources and atmospheric chemical processes, including secondary aerosol. The main components are water-insoluble and amino acid fluorophores in POA. However, the content of amino acid-like fluorophores in ambient PM is lower than that in POA, which suggest that the amino acid-like fluorophores have been photodegraded or transformed into HULIS fluorophores in ambient aerosol.

(2) Low oxidation HULIS could be converted into high oxidation HULIS in ambient PM. The result confirms the conclusion.

*4) In the paragraph starting at Line 274, the authors begin using the term 'light excitation' instead of photolysis or illumination which are used in the figures. It would improve the manuscript to make this terminology consistent. Also, in Line 280-281, what is meant by 'POA has certain oxidability'? Re-phrase whatever concept is trying to be conveyed here.*

We corrected "*light excitation*" to "*illumination*" in Sec 3.3 in improved paper.

We have deleted "*POA has certain oxidability*" to "*which suggest POA could generate $^1O_2$ without illumination*" in improved paper.

*5) The Implications section is still brief and lacks any reference to previous literature. This section could also be improved by broader interpretation of all the included results. For instance, how do the results in Sec. 3.3 on singlet oxygen generation connect with the results on degree of oxidation in Sec. 3.1 and 3.2, i.e. photolysis increases the degree of oxidation in the aerosol samples which in turn leads to a higher capacity for singlet oxygen formation via photosensitization reactions.*

We have revised Implications.

[revised manuscript text omitted]

*Minor Comments:*
*Line 2: Change 'photosensitiveness' to 'photosensitivity' and 'have' to 'has'*
We have corrected "*photosensitiveness*" to "*photosensitivity*" and "*have*" to "*has*".

*Line 10-13: Re-write this sentence: 'In terms of photochemical reactivity, the triplet state COM decreases slightly in ambient particulate matter samples but increases in primary organic aerosol (POA) following photolysis.*
We have corrected "*In terms of photochemical reactivity, compared with before photolysis, the triplet state COM (3COM*) decrease slightly in ambient particulate matter (ambient PM) samples, but increase in primary organic aerosol (POA)*" to "*COM Photodegradation has a significant impact on photochemical reactivity. The content of triplet state COM decreases slightly in ambient particulate matter but increases in primary organic aerosol following photodegradation*".

*Line 25: Change 'chemistry' to 'chemical'*

We have corrected "*secondary chemistry reactions*" to "*secondary aerosols*".

*Line 39-40: Change 'not complete clear' to 'unclear'*
We have deleted "*Yet the mechanisms of photo-bleaching process are still not complete clear*".

*Line 58: Change 'participate' to 'participates'*
We have corrected "*participate*" to "*participates*".

*Line 60: Change '' to 'solar irradiation'*
We have corrected "*Aromatic ketones could be excited to generate triplet state (3COM\*) under light conditions (Rosario-Ortiz and Canonica, 2016; Del Vecchio and Blough, 2004; Wenk et al., 2013; Ma et al., 2010)*" to "*such as aromatic ketones (Canonica et al., 2006; Marciniak et al., 1993), benzophenone (Encinas et al., 1985), and phenanthrene (Wawzonek & Laitinen, 1942), have been identified as the precursor of $^3COM*$*".

*Line 61: Change 'induce' to 'can generate'*
We have corrected "*$^3COM*$ induce reactive oxygen species (ROS)*" to "*COM participates in atmospheric photochemistry process indirectly through generating reactive intermediates*".

*Line 76: Change 'stated' to 'studied'*
We have corrected "*The effects of COM on photochemical reactivity and aerosol aging (photochemical reactivity is characterized by triplet state and singlet oxygen generation capacity) are also stated by the method of reactive species capture technology and electron paramagnetic resonance spectrometer (EPR)*" to "*to investigate the effects of COM photo-degradation on photochemical reactivity and aerosol aging (photochemical reactivity is characterized by triplet state and singlet oxygen generation capacity)*".

*Line 125: Change 'could refer to the previous literature' to 'has been described previously'*
We have corrected "*The method of organic carbon (OC) analysis could refer to the previous literature*" to "*The analytical method of carbonaceous components has been described previously*".

*Line 146 Change 'states' to 'state'*
We have corrected "*triplet states*" to "*$^3COM*$*".

*Line 229: Change 'study' to 'studied' and 'chromophores' to 'fluorophores' to highlight that you are referring to fluorescence here*
We have corrected "*Tang et al. (2020) study the chromophores in water-soluble and water-insoluble samples, respectively*" to "*Although previous study analyzed the water-soluble and water-insoluble fluorophores separately*".

*Line 230: Change ', respectively' to 'separately'*
We have corrected "*Tang et al. (2020) study the chromophores in water-soluble and water-insoluble samples, respectively*" to "*Although previous study analyzed the water-soluble and water-insoluble fluorophores separately*".

*Line 231: Change 'so that' to 'to'*
We have corrected "*so that*" to "*to*".

*Line 232: Change 'chromophores' to 'fluorophores'*
We have corrected "*chromophores*" to "*fluorophores*".

*Line 238: Rewrite as "The composition of the fluorophores changes significantly during the photolysis process."*
We have corrected "*The compositions of chromophores change significantly during photolysis process*" to "*The content of fluorophores changes significantly during the photodegradation process*".

*Line 260: Change 'states' to 'state'*
We have corrected "*states*" to "*state*".

*Line 263 Change 'not as expected' to 'unexpected'*
We have corrected "*not as expected*" to "*unexpected*".

*Line 274: Change to 'further induce singlet oxygen formation'*
We have corrected "*COM can generate triplet states and furtherly induce singlet oxygen*" to "*COM can generate triplet state and further generate singlet oxygen*".

*Line 278: Change 'is' to 'of'*
We have corrected "*is*" to "*of*".

---

## Author Response (AR3)

*Anonymous Referee #1 (reconsidered after major revisions)*

*The manuscript remains difficult to read due to grammatical errors, I started to correct them, but there are too many and this should have been checked by the authors. The discussion on water-soluble and insoluble fractions is still very confusing and the insoluble fraction is a methanolic extract. Was the methanolic extract irradiated? If so fundamental differences are expected due to the solvent. While the manuscript states that no significant changes between solvents were observed, this cannot be assumed during photodegradation experiments at all! What is the justification to call statistical components for the PARAFAC analysis low oxidation HULIS or high-oxidation HULIS? PARAFAC components have no chemical meaning and are solely derived from a statistical modelling. If such assumptions are made, they need to be defined by correlations with other analytical techniques that determine oxygenation levels or increases in oxygen-containing functional groups.*

We appreciate the comments from reviewer. According to the reviewer's comments, we have carefully revised this paper. Especially, we have further modified grammatical errors in the whole paper. The details changes are as follows. *The blue italics are comments of reviews. The red italics are improvements and original text of reviews.* The black font are responses.

1. We have corrected grammatical errors in the whole paper. For example,
(1) We have revised "*Photodegradation of precursors limit singlet oxygen generation and affect the aerosol photochemistry process. In conclusion, COM photodegradation not only change the compositions and properties, but also change aerosol aging*" to "*The combination of optical property, chemical component, and reactive oxygen species have an important impact on the atmosphere quality. The new insights on photodegradation of COM in aerosol reinforce the importance of studying DOM related with the photochemistry and aerosol aging*" in line 14-17 in the improved paper.
(2) We have revised "*[3]COM\* can generate reactive oxygen species (ROS)*" to "*[3]COM\* not only can produce photochemical reaction directly, but also can generate reactive oxygen species (ROS)*" in line 63-64 in the improved paper.
(3) We have revised "*Fig.4 show the difference of triplet state generation before and after the photodegradation*" to "*Fig.4 shows the variation of triplet state generation before and after the photodegradation*" in line 270-271 in the improved paper.
(4) The more changes are presented in the track-change-mode version of the manuscript. For example, L1, L5, L33, L43, L49, L66, L70, L80, L95, L126, L140, L169, L190, L209, L233, L304, L311, L320, L331, L356, L399, L408, and so on.

2. We have refined the extraction section in the improved paper.
Firstly, organic matter on the filter was extracted by water and the water-soluble organic matter (WSOM) was obtained. Secondly, residual organic matter on the filter was further extracted by methanol and we obtain the methanol-soluble organic matter (MSOM). Noted that only WSOM was used in the triplet state and singlet state experiment.
We have stated "*The samples with the photodegradation time of 0 and 7 d were defined as the original and photolyzed samples, respectively. Only WSOM of original and photolyzed samples was*

*used in the triplet state generation experiment. A capsule (Figure S2(c)) was designed for this*
*experiment*" in line 143-146 and "*Only WSOM of original and photolyzed was used in the singlet*
45  *oxygen generation experiment*" in line 162-163 in the improved paper.

3. EEMs could not reveal the chemical structure on the molecular level of complex fluorophores, but it can represent the overall fluorophores structure such as the chromophore types (Murphy et al., 2013; Chen et al., 2020, 2021a and b). The COM was identified as high-oxidation HULIS or low
50  oxidation HULIS, which is based on previous research by Chen et al (2016). They result suggests that these fluorophores have similar fluorescent functional groups and structures with high-oxidation or low oxidation ion fragments in HR-AMS. The result does not reveal that theses fluorophores must be high-oxidation or low oxidation organic compounds. EEM is a holistic approach to characterize complex COM in the environment.
55  We also stated that "*Pyrolysis carbon is identified as oxygen-containing organic substance. Thus, the increasing oxygen-containing organic matter may be due to the photo-inducing oxidation reaction*". The results suggest the effect of photooxidation on COM.
EEM is an important method for directly characterizing the occurrences, origins, and chemical behaviors of atmospheric chromophores. Examples are as follows.

[Figure]

60
EEM can identify the overall chemical structure of complex fluorophores (Graphical abstract, Chen et al., 2016).

[Figure]

EEM can identify the chemical structure and origins of complex fluorophores (Graphical abstract,
65  Chen et al., 2020).

[Figure]

EEM can identify the origins of complex fluorophores (Graphical abstract, Chen et al., 2021a).

[Figure]

EEM can tracer the degree of oxidation of complex fluorophores (Graphical abstract, Chen et al., 2021b).

➢ *Murphy, K. R., Stedmon, C. A., Graeber, D., et al.: Fluorescence spectroscopy and multi-way techniques. PARAFAC, Anal. Methods 2013, 5, 6557-6566.*

➢ *Chen, Q., Hua, X., Li, J., et al.: Identification of species and sources of atmospheric chromophores by fluorescence excitation-emission matrix with parallel factor analysis. Science of the total environment 2020, 718, 137322.*

➢ *Chen, Q., Hua, X., Li, J., et al.: Diurnal evolutions and sources of water-soluble chromophoric aerosols over Xi'an during haze event, in Northwest China. Science of the total environment 2021a, 786, 147412.*

➢ *Chen, Q., Hua, X., Dyussenova, A., et al.: Evolution of the chromophore aerosols and its driving factors in summertime Xi'an, Northwest China. Chemosphere 2021b, 281, 130838.*

➢ *Chen, Q., Miyazaki, Y., Kawamura, K., et al.: Characterization of chromophoric water-soluble organics in an urban, a forest, and the eastern equatorial Pacific aerosols using EEM and HR-ToF-AMS spectroscopy. Environ. Sci. Technol., 2016, 50, 10351-10360.*

*Some specific comments:*

*1. Line 14: precursors of what?*

We have revised "*Photodegradation of precursors limit singlet oxygen generation and affect the aerosol photochemistry process*" to "*which could be attributed to photodegradation of precursors of $^1O_2$*" in line 14 in the improved paper.

*2. Line 35: add "that" to "….due to chromophores that are photo-bleached in aerosols"*

We have revised "*due to chromophores are photo-bleached in aerosols*" to "*that due to COM is photo-bleaching in aerosol*" in line 36 in the improved paper.

*3. Line 36: secondary and not second*

We have revised "*second organic aerosol (SOA)*" to "*secondary organic aerosol (SOA)*" in line 40 in the improved paper.

*4. Line 37: define MAC*

We have defined "*mass absorption coefficents (MAC)*" in line 37 in the improved paper.
We have revised "*Lee et al. (2014) reported that the mass absorption coefficents (MAE)*" to "*Lee et al. (2014) also reported that the MAC*" in line 39 in the improved paper.

*5. Line 38: define OM*

We have revised "*wood-burning OM*" to "*wood-burning organic matter (OM)*" in line 38 in the

improved paper.

*6. Line 39: hydroxyl groups cannot be photodegraded. You might want to say hydroxylated aromatic phenols.*

We have revised "*hydroxyl groups*" to "*hydroxylated aromatic phenols*" in line 39 in the improved paper.

*7. Line 49: define POA*

We have revised "*POA*" to "*primary organic aerosol (POA)*" in line 49 in the improved paper.

*8. Line 52: comprehensively and not comprehensive*

We have revised "*comprehensive*" to "*comprehensively*" in line 52 in the improved paper.

*9. Line 55: What is the complex photochemical reaction?*

Extensive research has been done in COM photochemical reaction, such as generating reactive oxygen species, photo-Fenton reaction, and the formation of complexation of the metal ions with COM.

We have revised "*Atmospheric COM could participate in the complex photochemical reaction, which further affect the aerosol aging*" to "*Photochemical process of COM largely determines the aerosol aging*" in line 54 in the improved paper.

*10. Line 66: triplet state COM can also directly react and not just via ROS.*

We have revised "*$^3$COM\* can generate reactive oxygen species (ROS)*" to "*$^3$COM\* not only can produce photochemical reaction directly, but also can generate reactive oxygen species (ROS)*" in line 64 in the improved paper.

*11. Line 73-74: rephrase, this is grammatically confusing*

We have revised "*Why $^3$COM\* is employed not $^1$COM\*? The reasons are lower formation rate (15–100 times slower than $^1$COM\*), lower quenching rate (20000 times lower than $^1$COM\*), and highter steady-state concentrations of $^3$COM\* (200~1300 times higher than $^1$COM\*)*" to "*Compared with $^1$COM\*, the characteristics of $^3$COM\* are lower formation rate (15–100 times slower than $^1$COM\*), lower quenching rate (20000 times lower than $^1$COM\*), and higher steady-state concentration (200~1300 times higher than $^1$COM\*)*" in line 72-75 in the improved paper.

*12. Line 112: Please further define this fraction, while it is unsoluble in water it is still soluble in methanol. Water insoluble fraction could otherwise be mistaken for all the particulate COM, which is not the case. Why not calling it methanolic fraction after define it as being the fraction without the water-soluble components?*

Firstly, organic matter on the filter was extracted by water and the water-soluble organic matter was obtained. Secondly, residual organic matter on the filter was further extracted by methanol and we obtain the methanol-soluble organic matter.

We have revised "*water-insoluble organic matter (WISOM)*" to "*methanol-soluble organic matter (MSOM)*" in the improved paper.

 *13. Line 117. I still would briefly describe what was done and not solely referring to previous publications.*

We have described the method of OC/EC measurement in line 116-124 in the improved paper.

We added "*Organic carbon (OC) was measured in the absence of oxygen. An oven in the instrument was filled with helium and temperature was risen in a gradient style. Different temperatures are*
 *needed for particular analysis phases (OC1-310 ℃, OC2-472 ℃, OC3-615, OC4-850 ℃). Element carbon (EC) was measured in the present of oxygen. The oven in the instrument was filled with helium-oxygen gas mixture (He/$O_2$=10/1). Different temperatures were also needed for particular analysis phases (EC1-550 ℃, EC2-625 ℃, EC3-700℃, EC4-775 ℃, EC5-850 ℃, EC6-870 ℃). The products in the heating process were further oxidized to $CO_2$. The carbon content was obtained*
 *through the measurement of $CO_2$*" in line 116-124.

*14. Line 124: not exposure time. This is integration time.*

We have revised "*The exposure time was 0.5 s*" to "*The integration time was 0.5 s*" in line 130 the improved paper.

*15. Line 128: define PARAFAC and explain that a model was created from 111 samples. It sounds like you used somebody else's model but that is not the case, is it?*

We have revised "*The EEM data was analyzed by parallel factor analysis mode*" to "*The EEM data was analyzed by parallel factor analysis model (PARAFAC)*" in line 134 the improved paper.
 PARAFAC has already been used widely in identifying fluorophores and PARAFAC model has been described in previous study (Murphy et al., 2013). The program was public.

> ➢ *Murphy, K. R., Stedmon, C. A., Graeber, D., et al.: Fluorescence spectroscopy and multiway techniques. PARAFAC, Anal. Methods 2013, 5, 6557-6566.*

 *16. Line 132: You need to explain better why the 4-component model was selected and not any other.*
According to the EEM characteristics and the residual error variation trend of the 2-7 component PARAFAC models, 4 fluorescent components were identified. In the parallel factor analysis, the most appropriate number of components can be selected by judging the EEM characteristics and model residuals (Figure S4). In general, the smaller the sum of squared residuals and the ascensional
 range, the better the fitting effect, but increasing the number of components will add the difficulty of explanation. All things considered, the 4-component PARAFAC model is more appropriate.

*17. Line 183: I do not understand why it is claimed here that WSOC is completely photodegraded. Where is the evidence?*
 We have revised "*The EEM data was analyzed by parallel factor analysis mode*" to "*OM has been photodegraded adequately*" in line 193 the improved paper.

WSOC is nearly unchanged. POA is fresh and ambient PM has undergone long-term aerosol aging. Therefore, we stated that "*OM has been photodegraded adequately*" in real environment.

 *18. Line 236: PARAFAC components are statistically defined and hence do not have any chemical meaning. They are solely derived from the variations within a dataset. It is not at all clear how the oxidation level can be inferred on statistical components. On what kind of supporting data is this based?*

EEMs could not reveal the chemical structure on the molecular level of complex fluorophores, but it can represent the overall fluorophores structure such as the chromophore types (Murphy et al., 2013; Chen et al., 2020, 2021a and b). The COM was identified as high-oxidation HULIS or low oxidation HULIS, which is based on previous research by Chen et al (2016). They result suggests that these fluorophores have similar fluorescent functional groups and structures with high-oxidation or low oxidation ion fragments in HR-AMS. The result does not reveal that theses fluorophores must be high-oxidation or low oxidation organic compounds. EEM is a holistic approach to characterize complex COM in the environment.

➤ *Murphy, K. R., Stedmon, C. A., Graeber, D., et al.: Fluorescence spectroscopy and multiway techniques. PARAFAC, Anal. Methods 2013, 5, 6557-6566.*

➤ *Chen, Q., Hua, X., Li, J., et al.: Identification of species and sources of atmospheric chromophores by fluorescence excitation-emission matrix with parallel factor analysis. Science of the total environment 2020, 718, 137322.*

➤ *Chen, Q., Hua, X., Li, J., et al.: Diurnal evolutions and sources of water-soluble chromophoric aerosols over Xi'an during haze event, in Northwest China. Science of the total environment 2021a, 786, 147412.*

➤ *Chen, Q., Hua, X., Dyussenova, A., et al.: Evolution of the chromophore aerosols and its driving factors in summertime Xi'an, Northwest China. Chemosphere 2021b, 281, 130838.*

➤ *Chen, Q., Miyazaki, Y., Kawamura, K., et al.: Characterization of chromophoric water-soluble organics in an urban, a forest, and the eastern equatorial Pacific aerosols using EEM and HR-ToF-AMS spectroscopy. Environ. Sci. Technol., 2016, 50, 10351-10360.*

*Anonymous Referee #2 (accepted subject to minor revisions)*

*The manuscript 'Photodegradation of Atmospheric Chromophores: Changes in Oxidation State and Photochemical Reactivity' provides results on the photochemical aging of atmospheric aerosols (both ambient PM and laboratory generated POA). The results include OC/EC analysis, parallel factor (PARAFAC) analysis of excitation-emission matrices, and photosensitization of 1O2 with each measured as a function of solar irradiation. The manuscript has been improved sufficiently for publication. I only have minor comments to improve the readability of the manuscript. My comments are outlined below.*

220

We appreciate the positive comments from reviewer. According to the reviewer's comments, we have revised this paper. The details are as follows. *The blue italics are comments of reviews. The red italics are improvements and original text of reviews.* The black font are responses.

225

*Minor Comments:*
*1. Line 11: Change 'result' to 'results'*
We have revised "*result*" to "*results*" in line 12 in the improved paper.

230

*2. Line 15-16: Change 'change the compositions' to 'changes the composition'*
We have revised "*COM photodegradation not only change the compositions and properties*" to "*The combination of optical property, chemical component*" in line 14 in the improved paper.

235

*3. Line 25: Change 'originate' to 'originates'*
We have revised "*originate*" to "*originates*" in line 26 in the improved paper.

*4. Line 29: Change 'process' to 'processing'*
We have revised "*process*" to "*processing*" in line 31 in the improved paper.

240

*5. Line 34-35: Change 'chromophores are photo-bleached' to 'chromophore photo-bleaching'*
We have revised "*chromophores are photo-bleached*" to "*COM is photo-bleaching*" in line 37 in the improved paper.

245

*6. Line 49: Remove 'ability on'*
We have revised "*SOA may have a more significant ability on light absorption*" to "*SOA may have a more significant light absorption*" in line 48 in the improved paper.

250

*7. Line 55: Change 'the complex photochemical reaction' to 'complex photochemical reactions'*
We have revised "*Atmospheric COM could participate in the complex photochemical reaction, which further affect the aerosol aging*" to "*Photochemical process of COM largely determines the aerosol aging*" in line 54 in the improved paper.

255

*8. Line 63-64: 'deactivate quickly with the ways of' to 'deactivates by'*
We have revised "*$^1COM^*$ deactivate quickly with the ways of emitting photon (fluorescence) and intersystem crossing (triplet state, $^3COM^*$)*" to "*$^1COM^*$ deactivates by emitting photon (fluorescence) and intersystem crossing (triplet state ($^3COM^*$) generation)*" in line 62-63 in the

improved paper.

260

*9. Line 80: Change to 'in the laboratory'*
We have revised "*in laboratory*" to "*in the laboratory*" in line 81 in the improved paper.

*10. Line 132: Change 'Analysis error' to "Error analysis'*

265 We have revised "*Analysis error*" to "*Error analysis*" in line 138 in the improved paper.

*11. Line 182: Change 'opposite' to 'in contrast'*
We have revised "*opposite*" to "*in contrast*" in line 192 in the improved paper.

270 *12. Line 186: Change 'indicate' to 'indicates'*
We have revised "*indicate*" to "*indicates*" in line 197 in the improved paper.

*13. Line 187: Change 'opposite' to 'in contrast'*
We have revised "*opposite*" to "*in contrast*" in line 198 in the improved paper.

275

*14. Line 188-190: This sentence is unclear. Consider re-phrasing for clarity.*
We have revised "*Organic matter with high molecular weight is photocomposed to small molecular weight and the molecular weight tend to be consistent following the photodegradation*" to "*The proportion of different molecular weight OM is nearly unchanged following the photodegradation*

280 *in ambient PM*" in line 199-200 in the improved paper.

*15. Line 217: Change 'consider' to 'considered'*
We have revised "*consider*" to "*considered*" in line 228 in the improved paper.

285 *16. Line 259: Change 'promote' to 'promotes'*
We have revised "*promote*" to "*promotes*" in line 274 in the improved paper.

*17. Line 296: Change 'lead' to 'leads'*
We have revised "*lead*" to "*leads*" in line 315 in the improved paper.

290

*18. Line 307-308: Change '...chemical compositions, and photochemical activity. The characteristics of COM photo-degradation were revealed.' to '...chemical composition, and photochemical activity to reveal the characteristics of COM photo-degradation.'*
We have revised "*We made a comprehensive study in COM photo-degradation and the effect of*

295 *COM photo-degradation on optical properties, chemical compositions, and photochemical activity. The characteristics of COM photo-degradation were revealed*" to "*We made a comprehensive study in COM photodegradation and the effect of COM photodegradation on optical property, chemical component, and photochemical reactivity to reveal the characteristics of COM photodegradation*"
in line 321-323 in the improved paper.

300

*19. Line 320: Change 'dominant' to 'dominate'*
We have revised "*dominant*" to "*dominate*" in line 334 in the improved paper.

*20. Line 335: Change 'it' to 'this'*

We have revised "*There were two reasons for it*" to "*There were two reasons for this*" in line 350 in the improved paper.

*21. Line 339" Change 'would be" to 'could'*

We have revised "*COM photodegradation would be play an important role in the content of ROS*" to "*COM photodegradation could play an important role in the content of ROS*" in line 354 in the improved paper.

*22. Line 340: Change 'celebrate' to 'calibrate'. I believe this is the word that is intended.*

We have revised "*celebrate*" to "*calibrate*" in line 355 in the improved paper.

*23. Line 346: Change 'COM photodegradation have' to "COM photodegradation has a"*

We have revised "*COM photodegradation have different impact on*" to "*COM photodegradation had a different impact on*" in line 361 in the improved paper.

---

## Author Response (AR4)

*Editor Decision: Publish subject to technical corrections (acp-2020-1223)*

*Comments to the Author: Dear authors. I have examined the final version of the manuscript, and while I think it could be further improved with more precise wording choices, I do not think another round of review is going to help. The results are certainly useful, and deserve to be published, so I am going to go ahead with the acceptance. The manuscript still has an unusually large number of typos, grammatical mistakes, and imprecise wording choices. For example, below is a partial list from the first two paragraphs in the paper. I am going to ask Copernicus (ACP publisher) about their copy-editing services but it is better if they start with a cleaner version. Therefore, please go through the text one more time, perhaps enlisting your own professional editing services.*

We appreciate the positive comments from editor. According to the editor's comments, we have revised this paper. The details are as follows. *The blue italics are comments of reviews. The red italics are improvements and original text.* The black font are responses.

*1. L24: mainly originates from -> is an important component of*
We have revised "*mainly originates from*" to "*is an important component of*" in line 23 in the improved paper.

*2. L25: emission -> emissions*
We have revised "*emission*" to "*emissions*" in line 24 in the improved paper.

*3. L27: absorption for -> absorption of*
We have revised "*Because of the significant absorption for short-wave radiation (Wavelength range from near-ultraviolet light to visible light)*" to "*COM has a significant absorption in the near-ultraviolet and visible region*" in line 25-26 in the improved paper.

*4. L27: (Wavelength -> (wavelength*
We have revised "*Because of the significant absorption for short-wave radiation (Wavelength range from near-ultraviolet light to visible light)*" to "*COM has a significant absorption in the near-ultraviolet and visible region*" in line 25-26 in the improved paper.

*5. L29: atmospheric components and quality -> air quality*
We have revised "*atmospheric components and quality*" to "*air quality*" in line 27 in the improved paper.

*6. L32: As photosensitive substances, the -> The*
We have deleted "*As photosensitive substances, the optical properties and components of COM change significantly under solar irradiation*" in the improved paper.

*7. L34: that due to COM is -> due to*
We have revised "*On the one hand, optical properties change significantly that due to COM is photo-bleaching in aerosol*" to "*Photodegradation changes the optical property of COM*" in line 30 in the improved paper.

*8. L36: because -> when*

We have revised "*because*" to "*when*" in line 32 in the improved paper.

*9. L37: the sentence needs to be broken in two sentences*

We have revised "*Zhong and Jang (2014) reported that mass absorption coefficents (MAC) decreased by 41% on average because wood-burning organic matter (OM) was bleaching, such as conjugated aromatic rings and phenols, and hydroxylated aromatic phenols*" to "*Zhong and Jang (2014) reported that mass absorption coefficents (MAC) of wood-burning organic matter (OM) decreased by 41% on average. Conjugated aromatic rings, phenols, and hydroxylated aromatic phenols were the main components in wood-burning OM and these components were photo-bleaching*" in line 31-34 in the improved paper.

*10. L39: spectral -> spectral range*

We have revised "*spectral*" to "*spectral range*" in line 35 in the improved paper.

*11. L43: delete "could be"*

We have deleted "*could be*" in line 40 in the improved paper.

*... and so on – there are too many mistakes to list here.*

**We also corrected other mistakes in the improved paper. For example,**

1. We have revised "*COM can be decomposed into small molecules after photodegradation and the photodegraded COM may have lower volatility and higher oxidation degree*" to "*photodegradation could cause photochemical decomposition of COM and the decomposed COM is characterized by smaller molecule weight, lower volatility, and higher oxidation degree*" in line 37-38 in the improved paper.

2. We have revised "*COM photochemistry may dominate the chemical composition and the aerosol aging process*" to "*Photodegradation may dominate the chemical component of COM and aerosol aging*" in line 72 in the improved paper.

3. We have revised "*Straw and coal burning were the main way of heating and cooking in the rural areas in China*" to "*The main ways of heating and cooking were straw and coal burning in rural China*" in line 88 in the improved paper.

4. We have revised "*The original and photodegraded samples were ultrasonic extracted with ultrapure water (>18.2 MΩ•cm, Hitech, China) and filtered through a 0.45 μm filter (Jinteng, China) to obtain the water-soluble organic matter (WSOM)*" to "*Water-soluble organic matter (WSOM) was extracted from the original and photodegraded samples by sonication in ultrapure water (>18.2 MΩ•cm, Hitech, China) and filtered through a 0.45 μm filter (Jinteng, China)*" in line 97-99 in the improved paper.

5. We have revised "*temperature was risen in a gradient style. Different temperatures are needed for particular analysis phases*" to "*Temperature of the oven risen and the different phases were at a selected temperature*" in line 107 in the improved paper.

6. We have revised "*As short-lived reactive intermediates, 3COM\* has an important impact on photochemical process in atmospheric environment*" to "*$^{3}$COM\* is a short-lived reactive intermediate and has an important impact on photochemical process in aerosol*" in line 129-130 in the improved paper.

7. We have revised "*TMP was used as the capturing agent for the $^{3}$COM\**" to "*TMP was the capturing agent for $^{3}$COM\**" in line 134 in the improved paper.

8. We have revised "*Phenol solution was used as the internal standard substance for TMP quantification*" to "*Phenol was the interior label for TMP quantification*" in line 140 in the improved paper.

9. We have revised "*COM can be decomposed and transformed due to photodegradation in aerosol*" to "*Photodegradation causes the decomposition and transformation of fluorophores*" in line 223 in the improved paper.

10. We have added "*higher*" in line 287 in the improved paper.

The more changes are presented in the track-change-mode version of the manuscript. For example, line 255-260, line 329-330, and so on.